# Improved Distribution Estimation in $\ell_\infty$

**Doron Cohen** [1]   **Aryeh Kontorovich** [1]   **Yonatan Livshitz** [1]

## Abstract

We present improved bounds for estimating discrete probability distributions under the $\ell_\infty$ norm. These include minimax bounds in expectation and high-probability tail bounds. We resolve some of the open questions posed in Kontorovich and Painsky (JMLR, 2025) — including a fully empirical version of the tightest risk bound they presented and identifying the form of the worst-case extremal distribution. Encouraging empirical results are reported as well.

## 1. Introduction

Estimating an unknown discrete distribution from i.i.d. samples is a classical problem, with a long history of sharp minimax theory under $\ell_1$ and $\ell_2$ losses. In a range of modern applications—including uniform calibration, anomaly detection, and goodness-of-fit pipelines that trigger on the largest coordinate-wise discrepancy—the relevant performance metric is instead the sup-norm loss

$$\|\hat{p} - p\|_\infty = \sup_{i \geq 1} |\hat{p}_i - p_i|,$$

where $p$ is a probability distribution over a (possibly infinite) countable alphabet and $\hat{p}$ is an estimator based on $n$ samples.

The $\ell_\infty$ loss is, in a sense, simple: an $O(1/\sqrt{n})$ risk decay holds uniformly over all discrete distributions. The technical challenge is therefore to obtain delicate, distribution-dependent *fast rates* and fully empirical counterparts whose sharpness reflects that the effective difficulty is governed by the unknown tail profile of $p$, rather than merely by its support size.

Motivated by various problems in applied statistics, a recent work of Kontorovich & Painsky (2025) initiated the nonasymptotic study of $\ell_\infty$ distribution estimation in the countable-alphabet regime. In particular, they introduced two distribution-dependent complexity proxies that capture the interplay between variance and tail decay:

$$
\begin{aligned}
v^*(p) &:= \sup_{i \geq 1} p_i(1 - p_i), \\
V^*(p) &:= \sup_{i \geq 1} p_i^\downarrow (1 - p_i^\downarrow) \log(i + 1)
\end{aligned}
$$

(where $p^\downarrow$ is the nonincreasing permutation) and established upper bounds (in expectation and high probability) for the MLE $\hat{p}$ in terms of $v^*(p)$ and $V^*(p)$. They also raised several open questions, including (i) whether one can obtain *fully empirical* (data-dependent) confidence bounds without unknown functionals of $p$, and (ii) what is the true *least-favorable* distribution governing the worst-case risk.

In this paper we sharpen and extend the theory in both directions. Our results give improved minimax bounds (in expectation and deviation form), identify a simple extremal distribution that is worst-case up to universal constants and can be chosen independently of $n$, and provide a tight *fully empirical* high-probability guarantee that depends only on the observed sample through natural plug-in statistics. We also report empirical evidence demonstrating that these bounds are informative in finite samples.

**Contributions (informal).**

- **Minimax and least-favorable distribution.** We show that the worst-case expected $\ell_\infty$ error scales as $\Theta(n^{-1/2})$, and that a two-point distribution $p^\star = (\frac{1}{2}, \frac{1}{2}, 0, \dots)$ is least-favorable up to a universal constant factor, uniformly for all sufficiently large $n$.

- **Fully empirical confidence bound.** We prove a high-probability inequality for $\|\hat{p} - p\|_\infty$ that is *fully empirical*, depending on $p$ only through plug-in quantities $\hat{v}^*$ and an empirical analogue $\hat{V}^*$ of $V^*(p)$.

- **Refined distribution-dependent regime picture.** Using sharp local Glivenko–Cantelli bounds, we relate the estimation error to two explicit complexity terms that exhibit different tail behaviors and quantify when each one dominates.

[1]Ben-Gurion University of the Negev, Beer-Sheva, Israel. Correspondence to: Doron Cohen <doronv@post.bgu.ac.il>, Aryeh Kontorovich <karyeh@bgu.ac.il>, Yonatan Livshitz <livshitz@post.bgu.ac.il>.

*Proceedings of the 43$^{rd}$ International Conference on Machine Learning*, Seoul, South Korea. PMLR 306, 2026. Copyright 2026 by the author(s).

# 2. Main Results

**Setting.** Let $p = (p_i)_{i \geq 1}$ be a distribution on $\mathbb{N}$ and let $X^n = (X_1, \ldots, X_n)$ be i.i.d. from $p$. Let $c_i(X^n) := \sum_{t=1}^{n} \mathbf{1}\{X_t = i\}$ and denote the empirical pmf (MLE) by $\hat{p}_i := c_i(X^n)/n$. We study the sup-norm error $\|\hat{p} - p\|_\infty$ and its expectation

$$\Delta_n(p) := \mathbb{E}_p[\|\hat{p} - p\|_\infty].$$

Write $p^\downarrow$ for $p$ sorted in nonincreasing order, and define

$$v_i(p) := p_i(1 - p_i),$$
$$v^*(p) := \sup_{i \geq 1} v_i(p),$$
$$V^*(p) := \sup_{i \geq 1} v_i(p^\downarrow) \log(i + 1).$$

We also define empirical analogues

$$\hat{v}^* := \sup_{i \geq 1} \hat{p}_i(1 - \hat{p}_i),$$
$$\hat{V}^* := \sup_{i \geq 1} \hat{p}_{[i]}(1 - \hat{p}_{[i]}) \log(i + 1),$$

where $\hat{p}_{[i]}$ are the order statistics of $(\hat{p}_i)_{i \geq 1}$.

We write $f \lesssim g$ if $f \leq cg$ where $c$ is an absolute constant.

## 2.1. Minimax behavior and a fixed least-favorable distribution

Our first result resolves the extremal structure of the worst-case expected $\ell_\infty$ risk, answering an open question of Kontorovich & Painsky (2025).

**Theorem 2.1** (A fixed least-favorable distribution; minimax rate). *There exist universal constants $c > 0$, $C > 0$, and $n_0 \in \mathbb{N}$ such that the following holds for all $n \geq n_0$. Let $p^\star := (\frac{1}{2}, \frac{1}{2}, 0, 0, \ldots)$. Then*

$$\Delta_n(p^\star) \geq c\,\Delta_n(p) \qquad \text{for all distributions } p \text{ on } \mathbb{N},$$

*and consequently*

$$c\,n^{-1/2} \leq \sup_p \Delta_n(p) \leq C\,n^{-1/2}.$$

**Remark.** Theorem 2.1 shows that, up to universal constants, the hardest instance for $\ell_\infty$ estimation is already present on a two-point alphabet, and does *not* require an $n$-dependent effective support size. In particular, the global minimax rate in expectation is $\Theta(n^{-1/2})$, with a least-favorable $p^\star$ independent of $n$.

## 2.2. A fully empirical high-probability bound

Next we give a deviation inequality for $\|\hat{p} - p\|_\infty$ that is *fully empirical*, replacing the unknown functionals $V^*(p), v^*(p)$ by observable plug-in quantities.

**Theorem 2.2** (Fully empirical $\ell_\infty$ bound). *There exist universal constants $c_1, c_2 > 0$ such that for all $\delta \in (0, 1)$, with probability at least $1 - 2\delta - \frac{162}{n}$,*

$$\|\hat{p} - p\|_\infty \leq c_1 \sqrt{\frac{\hat{V}^*}{n} + \frac{\hat{v}^*}{n} \log \frac{1}{\delta}}$$
$$+ c_2 \left( \frac{\log(n/\delta)}{n} + \frac{\log n}{n} \right).$$

*Remark* 2.3. The probability lower bound can be written in a standard $1 - 2\bar{\delta}$ form by setting $\bar{\delta} := \delta + 81/n$, since $1 - 2\delta - 162/n = 1 - 2\bar{\delta}$. In typical usage $n$ is large and one may take $\delta \gtrsim 1/n$ so that the extra term is absorbed.

**Remark.** The bound in Theorem 2.2 adapts automatically to the unknown tail profile of $p$ through $\hat{V}^*$, while retaining the correct $\sqrt{\log(1/\delta)/n}$ behavior in the worst case (e.g. under $p^\star$ from Theorem 2.1). Unlike bounds expressed in terms of $V^*(p)$, it can be evaluated directly from data.

## 2.3. Refined distribution-dependent bounds via local Glivenko–Cantelli

To connect with sharp distribution-dependent behavior beyond the global minimax regime, we also consider the functionals

$$S(p) := \sup_{j \geq 1} p_j^\downarrow \log(j + 1),$$
$$M(p) := \sup_{j \geq 1} \frac{\log(j + 1)}{\log\left(2 + \frac{\log(j+1)}{n\, p_j^\downarrow}\right)}.$$

These quantities arise naturally in sharp local Glivenko–Cantelli theory (cf. Cohen & Kontorovich, 2023; Blanchard & Voráček, 2024; Cohen et al., 2025) and yield a complementary "regime picture" for $\|\hat{p} - p\|_\infty$.

**Theorem 2.4** (High-probability bound via LGC functionals). *There exists a universal constant $C > 0$ such that for all $\delta \in (0, e^{-2}]$, with probability at least $1 - 2\delta$,*

$$\|\hat{p} - p\|_\infty \leq C \left( \sqrt{\frac{S(p)}{n}} \vee \frac{M(p)}{n} \right) \log \frac{1}{\delta}.$$

**Lemma 2.5** (Size of $M(p)$ and examples). *There are universal constants $C_1, c_1 > 0$ such that, for every distribution $p$ on $\mathbb{N}$ and every $n \in \mathbb{N}$,*

$$c_1 < M(p) < C_1 \log n.$$

*Moreover, there are universal constants $C_2, c_2 > 0$ and there exist distributions $p_1, p_2$ on $\mathbb{N}$ such that for all $n \in \mathbb{N}$,*

$$M(p_1) > C_2 \log n, \qquad M(p_2) < c_2.$$

**Lemma 2.6** (Size of $S(p)$). *There is a constant $c > 0$ such that for every distribution $p$ on $\mathbb{N}$,*

$$S(p) \; < \; c.$$

*Moreover, for every $\varepsilon > 0$, there exists a distribution $p$ on $\mathbb{N}$ such that*

$$S(p) \; < \; \varepsilon.$$

**Remark.** Theorem 2.4 shows that the distribution-dependent behavior of $\|\widehat{p}-p\|_\infty$ is governed by two different features of the ordered mass profile $p^\downarrow$. The term $S(p)$ measures the largest local variance block: at rank $j$, the quantity $p_j^\downarrow \log(j+1)$ is the cost of simultaneously controlling roughly the first $j$ coordinates when each has mass at least $p_j^\downarrow$. Thus $\sqrt{S(p)/n}$ is the sub-Gaussian part of the error. It dominates when some block of coordinates carries enough probability mass for ordinary variance fluctuations to be the main source of error. The term $M(p)$ captures a different phenomenon. It is sensitive to how many low-probability symbols are still visible at sample size $n$. Such symbols may each have very small variance, but because there can be many of them, the maximum empirical fluctuation can be driven by rare counts in the tail. This is the sub-gamma, or Poissonian, part of the local Glivenko–Cantelli behavior.

The examples in Lemmas 2.5–2.6 illustrate the range of possibilities. For a very concentrated distribution such as $(1/2, 1/2, 0, \dots)$, the tail term $M(p)$ remains only of constant order, and the usual $n^{-1/2}$ variance behavior is the relevant one. For other profiles, such as the polynomial-tail example used in the proof of Lemma 2.5, $M(p)$ can be as large as order $\log n$, showing that the tail contribution can be genuinely non-negligible. On the other hand, if the mass is spread over a very large effective alphabet, as in a uniform distribution on $A$ symbols with large $A$, then $S(p) = \log(A+1)/A$ becomes very small, while $M(p)$ stays in its constant-scale regime. In that case both contributions $\sqrt{S(p)/n}$ and $M(p)/n$ are small. Thus the theorem captures three qualitatively different situations: variance domination, tail domination, and highly spread-out profiles where both mechanisms lead to a small error.

## 3. Related Work

A sharp nonasymptotic study of $\ell_\infty$ distribution estimation over countable alphabets was initiated by Kontorovich & Painsky (2025); see also the works cited therein. They introduced the distribution-dependent complexity proxies $v^*(p)$ and $V^*(p)$, and derived nearly optimal expectation and high-probability bounds for the MLE in terms of these quantities, along with partially empirical counterparts. Our refined distribution-dependent bounds also draw on the recently developed *local Glivenko–Cantelli* framework, which seeks dimension-free, distribution-dependent

uniform convergence rates beyond the classical worst-case $\sqrt{\log d/n}$ regime. Cohen & Kontorovich (2023) established foundational characterizations and sharp rates for LGC in the product-measure setting, and subsequent works provided tighter regime descriptions and minimax optimality statements for the empirical mean estimator (Blanchard & Voráček, 2024; Blanchard et al., 2024; Cohen et al., 2025).

## 4. Proofs

### 4.1. Proof of Theorem 2.1

*Proof of Theorem 2.1.* We prove the theorem by reducing the multinomial (dependent) sampling model to an independent Bernoulli model via two decoupling inequalities. For the independent model, sharp characterizations of the expected $\ell_\infty$ error are available from Blanchard & Voráček (2024). We then maximize these tight bounds over all admissible distributions to identify the least-favorable distribution $p^\star$.

Let $X^{(1)}, \dots, X^{(n)}$ be independent and identically distributed random vectors with

$$X^{(j)} \sim \mathrm{Multinomial}(1, \mathbf{p}), \qquad j = 1, \dots, n,$$

where $\mathbf{p} = (p_1, p_2, \dots)$ is a probability vector on $\mathbb{N}$. For each coordinate $i \geq 1$, define the centered empirical deviations

$$Z_i := \frac{1}{n} \sum_{j=1}^{n} X_i^{(j)} - p_i, \qquad Z_i' := p_i - \frac{1}{n} \sum_{j=1}^{n} X_i^{(j)},$$

where $X_i^{(j)}$ denotes the $i$th coordinate of $X^{(j)}$.

Define also $\tilde{Z}_i, \tilde{Z}_i'$ be independent random variables with the same marginal distributions as $Z_i$ and $Z_i'$, respectively. Define $\tilde{\Delta}_n(p) := \mathbb{E} \max(\max_i \tilde{Z}_i, \max_i \tilde{Z}_i')$.

$$\mathbb{E}\|\hat{\boldsymbol{p}} - \boldsymbol{p}\|_\infty = \mathbb{E} \max_i \left| \frac{1}{n} \sum_{j \in [n]} X_{i,j} - p_i \right|$$
$$= \mathbb{E}\left[ \max\left( \max_i Z_i, \max_i Z_i' \right) \right].$$

Combining Corollary 3 in Blanchard et al. (2024) and A.1,

$$\mathbb{E} \max_i Z_i \geq \frac{1}{2} \mathbb{E} \max_i \tilde{Z}_i, \quad \mathbb{E} \max_i Z_i' \geq \frac{1}{2} \mathbb{E} \max_i \tilde{Z}_i'$$

Thus,

$$\mathbb{E} \max(\max_i Z_i, \max_i Z_i') \geq \mathbb{E} \frac{\max_i Z_i + \max_i Z_i'}{2}$$
$$\geq \frac{\mathbb{E} \max_i \tilde{Z}_i + \mathbb{E} \max_i \tilde{Z}_i'}{4}$$

$$\frac{\mathbb{E}\max_i \tilde{Z}_i + \mathbb{E}\max_i \tilde{Z}_i'}{4} \geq \frac{1}{4}\mathbb{E}\max(\max_i \tilde{Z}_i, \max_i \tilde{Z}_i')$$
$$= \frac{1}{4}\tilde{\Delta}_n(p).$$

By Kontorovich (2023, Proposition 3),

$$\frac{e}{e-1}\tilde{\Delta}_n(p) \geq \mathbb{E}\|\hat{\boldsymbol{p}} - \boldsymbol{p}\|_\infty.$$

Thus we have:

$$\frac{e}{e-1}\tilde{\Delta}_n(p) \geq \mathbb{E}\|\hat{\boldsymbol{p}} - \boldsymbol{p}\|_\infty \geq \frac{1}{4}\tilde{\Delta}_n(p)$$

For the independent Bernoulli model, Blanchard & Voráček (2024) proved that for $\mathbf{p} \in [0, 1/2]^{\mathbb{N}}_{\downarrow 0}$ and $\hat{\boldsymbol{p}} = (\frac{1}{n}\sum_{j=1}^n \tilde{X}_{1j}, \dots)$ there exist universal constants $c_1, c_2 > 0$ such that

$$c_1\left(1 \wedge \left(\sqrt{\frac{S(p)}{n}} \vee \frac{M(p)}{n}\right)\right) \leq \tilde{\Delta}_n(p)$$
$$\tilde{\Delta}_n(p) \leq c_2\left(1 \wedge \left(\sqrt{\frac{S(p)}{n}} \vee \frac{M(p)}{n}\right)\right).$$

Note that the restriction $\mathbf{p} \in [0, 1/2]^{\mathbb{N}}_{\downarrow 0}$ is justified. Indeed, define $\mathbf{p}'$ such that for all $i$

$$p_i' = \begin{cases} 1 - p_i & p_i > \frac{1}{2}, \\ p_i, & p_i \leq \frac{1}{2}. \end{cases}$$

Because of the symmetry of bernuli $\tilde{\Delta}_n(\mathbf{p}) = \tilde{\Delta}_n(\mathbf{p}')$. Because of $\sum_i p_i = 1$ the only index of $\mathbf{p}$ that can exceed $\frac{1}{2}$ is $p_1$ and thus $\sum_i p_i' \leq 1$ holds. From now on we will treat every distribution $\mathbf{p}$ as if it undergone the transformation to $\mathbf{p}'$ and thus use the assumptions $\mathbf{p} \in [0, 1/2]^{\mathbb{N}}_{\downarrow 0}$, $\sum_i p_i \leq 1$.

Thus, the following holds for some constants $C_1, C_2$:

$$C_1\left(1 \wedge \left(\sqrt{\frac{S(p)}{n}} \vee \frac{M(p)}{n}\right)\right) \leq \mathbb{E}\|\hat{\boldsymbol{p}} - \boldsymbol{p}\|_\infty$$
$$\mathbb{E}\|\hat{\boldsymbol{p}} - \boldsymbol{p}\|_\infty \leq C_2\left(1 \wedge \left(\sqrt{\frac{S(p)}{n}} \vee \frac{M(p)}{n}\right)\right).$$

Since $\mathbb{E}\|\hat{p} - p\|_\infty$ is bounded above and below by universal constant multiples of the objective in 1, to find $p^*$ it suffices to characterize the maximizers of this objective.

$$\sup_{j \geq 1}\left(\sqrt{\frac{p_j \log(j+1)}{n}} \vee \frac{\log(j+1)}{n\log\left(2 + \frac{\log(j+1)}{np_j}\right)}\right). \quad (1)$$

Let $p$ be any distribution and let $j^\star$ attain the supremum in 1. Because $p$ is sorted in non-increasing order and $\sum_i p_i \leq 1$, we have $p_{j^\star} \leq 1/j^\star$. Furthermore, for fixed $j$, both terms in 1 are monotone increasing functions of $p_j$.

Define $U_{j^\star}$ as the uniform distribution over $\{1, \dots, j^\star\}$. Then $U_{j^\star} = 1/j^\star \geq p_{j^\star}$, and therefore

$$\left(\sqrt{\frac{U_{j^\star,j^\star}\log(j^\star+1)}{n}} \vee \frac{\log(j^\star+1)}{n\log\left(2 + \frac{\log(j^\star+1)}{nU_{j^\star,j^\star}}\right)}\right) \geq$$
$$\geq \left(\sqrt{\frac{p_{j^\star}\log(j^\star+1)}{n}} \vee \frac{\log(j^\star+1)}{n\log\left(2 + \frac{\log(j^\star+1)}{np_{j^\star}}\right)}\right).$$

Hence the supremum over all distributions is attained by some uniform distribution on a finite support. Consequently, the search for extremal distributions may be restricted to the class $\{U_j : j \geq 2\}$. Note that $U_1$ is not relevant because $p \in [0, 1/2]^{\mathbb{N}}_{\downarrow 0}$.

It remains to determine the dimension of this uniform distribution, that is finding the following value

$$\arg\max_{j \geq 2}\left(\sqrt{\frac{\log(j+1)}{jn}} \vee \frac{\log(j+1)}{n\log\left(2 + \frac{j\log(j+1)}{n}\right)}\right).$$

Rewritten

$$\sup_{j \geq 2}\left(\sqrt{\frac{\log(j+1)}{jn}}\right) \vee \sup_{j \geq 2}\left(\frac{\log(j+1)}{n\log\left(2 + \frac{j\log(j+1)}{n}\right)}\right).$$

Because $\arg\max_{j \geq 2}\frac{1}{j}\log(j+1) = 2$ this becomes

$$\sqrt{\frac{\log 3}{2} \cdot \frac{1}{n}} \vee \frac{1}{n}\sup_{j \geq 1}\left(\frac{\log(j+1)}{\log\left(2 + \frac{\log(j+1)}{np_j}\right)}\right).$$

From 2.5 we know that for some constant $C$ and for all $n$, $C\log n \geq M(p)$. Thus, because of the faster decay in $n$ there exists $n_0$ for which for all $n \geq n_0$, $\sqrt{\frac{\log 3}{2} \cdot \frac{1}{n}} > \frac{M(p)}{n}$. Thus for all $n \geq n_0$

$$\arg\max_{j \geq 1}\left(\sqrt{\frac{\log(j+1)}{jn}} \vee \frac{\log(j+1)}{n\log\left(2 + \frac{j\log(j+1)}{n}\right)}\right) = 2$$

and $p^* = (\frac{1}{2}, \frac{1}{2}, 0, \dots)$. $\qquad\square$

*Proof of Theorem 2.2.* By Theorem 3 in Kontorovich & Painsky (2025), we have that with probability $\geq 1 - \delta - \frac{81}{n}$,

$$\|p - \hat{p}\|_\infty \leq 2\sqrt{\frac{V^*}{n} + \frac{v^*}{n}\log\frac{2}{\delta}}$$
$$+ \frac{4}{3n}\log\frac{2(n+1)}{\delta} + \frac{\log n}{n}.$$

It follows that

$$\|p - \hat{p}\|_\infty \leq 2\sqrt{\frac{\hat{V}^*}{n} + \frac{\hat{v}^*}{n}\log\frac{2}{\delta}}$$

$$+ \sqrt{\frac{1}{n}}\sqrt{\left|\hat{V}^* - V^*\right| + \left|\hat{v}^* - v^*\right|\log\frac{2}{\delta}}$$

$$+ \frac{4}{3n}\log\frac{2(n+1)}{\delta} + \frac{\log n}{n}.$$

Denote the bound in Kontorovich & Painsky (2025, Lemma 4) by

$$f(n,\delta) = a + 3b^2/2 + b\sqrt{a} + 3b\sqrt{\hat{v}^*}/2,$$

where

$$a = \frac{4}{3n}\log\frac{2(n+1)}{\delta} + \frac{\log n}{n},$$

$$b = 2\sqrt{\frac{\log(n+1)}{n} + \frac{1}{n}\log\frac{2}{\delta}}.$$

From Lemma A.2 we get

$$\left|\hat{V}^* - V^*\right| \leq \|p - \hat{p}\|_\infty \log\frac{1}{\|p - \hat{p}\|_\infty},$$

and from A.3,

$$\|p - \hat{p}\|_\infty \leq 2\sqrt{\frac{\hat{V}^*}{n} + \frac{\hat{v}^*}{n}\log\frac{2}{\delta}}$$

$$+ \sqrt{\frac{f(n,\delta)}{n}\log\frac{6n}{\delta}} + \frac{4}{3n}\log\frac{2(n+1)}{\delta}$$

$$+ \frac{\log n}{n}$$

with probability $\geq 1 - 2\delta - \frac{162}{n}$. Using the definitions of $a$ and $b$ from Lemma A.3, we have

$$f(n,\delta) \lesssim \frac{1}{n}\log\frac{n}{\delta} + \sqrt{\frac{\hat{v}^*}{n}\log\frac{n}{\delta}}.$$

The cross term in the expansion of $\|\hat{p} - p\|_\infty$ involves

$$T := \sqrt{\frac{f(n,\delta)}{n}\log\frac{6n}{\delta}}$$

$$\lesssim \frac{1}{n}\log\frac{n}{\delta} + \left(\frac{\hat{v}^*}{n}\right)^{1/4}\frac{(\log(n/\delta))^{3/4}}{\sqrt{n}}.$$

By the AM-GM inequality $\sqrt{xy} \leq (x+y)/2$ with $x = \sqrt{\frac{\hat{v}^*}{n}\log(n/\delta)}$ and $y = \frac{1}{n}\log(n/\delta)$, the second term satisfies

$$\left(\frac{\hat{v}^*}{n}\right)^{1/4}\frac{(\log(n/\delta))^{3/4}}{\sqrt{n}} = \sqrt{xy}$$

$$\leq \frac{1}{2}\sqrt{\frac{\hat{v}^*}{n}\log\frac{n}{\delta}} + \frac{1}{2n}\log\frac{n}{\delta}.$$

Thus, $T$ can be absorbed into the main variance term and the $O(1/n)$ term by adjusting the universal constants. Putting everything together,

$$\|p - \hat{p}\|_\infty \leq c_1\sqrt{\frac{\hat{V}^*}{n} + \frac{\hat{v}^*}{n}\log\frac{1}{\delta}}$$

$$+ c_2\left(\frac{\log(n/\delta)}{n} + \frac{\log n}{n}\right).$$

$\square$

## 4.2. Minimax result with $V^*$ constraint

**Proposition 4.1** (Minimax lower bound). *Let $n \geq 2$ and let $V' \in (0, \frac{1}{4}\log 2]$. Then for any estimator $\tilde{p} = \tilde{p}(X^n)$, there exists a distribution $p$ supported on $\mathbb{N}$ such that $V^*(p) \leq V'$ and*

$$\mathbb{E}_p\left[\|\tilde{p} - p\|_\infty\right] \geq c\sqrt{\frac{V'}{n}},$$

*where $c > 0$ is a universal constant.*

*Proof.* We use Le Cam's method with two hypotheses $p^{(1)}$ and $p^{(2)}$ constructed to satisfy the constraint $V^*(p^{(j)}) \leq V'$ while remaining difficult to distinguish.

**Construction of Hypotheses.** Let $\theta = \frac{V'}{2\log 2}$. Since $V' \leq \frac{1}{4}\log 2$, we have $\theta \leq \frac{1}{8}$. Let $\varepsilon \in (0, \theta)$. Define two distributions on the support $\{1, 2, 3, 4, \dots\}$:

$$p^{(1)} = (\theta + \varepsilon, \theta - \varepsilon, 1 - 2\theta, 0, \dots),$$

$$p^{(2)} = (\theta - \varepsilon, \theta + \varepsilon, 1 - 2\theta, 0, \dots).$$

We first verify the constraint $V^*(p^{(j)}) \leq V'$. The functional is defined as $V^*(p) = \sup_{k\geq 1} p_{(k)}(1 - p_{(k)})\log(k+1)$, where $p_{(k)}$ denotes the $k$-th largest probability. Since $\theta \leq \frac{1}{8}$ and $\varepsilon < \theta$, the mass $\theta + \varepsilon < \frac{1}{4}$ is at least $1 - 2\theta > \frac{3}{4}$, making it strictly larger than $\theta + \varepsilon$. Thus, for both hypotheses, the sorted probability vector $p^{(j)\downarrow}$ is:

$$p^{(j)\downarrow} = (1 - 2\theta, \theta + \varepsilon, \theta - \varepsilon, 0, \dots).$$

We evaluate the term $v_k\log(k+1) = p_{(k)}^{(j)\downarrow}(1 - p_{(k)}^{(j)\downarrow})\log(k+1)$ for each index $k$:

- $k = 1$: The mass is $1 - 2\theta$.

  $$v_1\log 2 = (1 - 2\theta)(2\theta)\log 2 = 2\theta(1 - 2\theta)\log 2$$
  $$\leq 2\theta\log 2 = V'.$$

  The inequality holds because $1 - 2\theta < 1$. By our choice of $\theta$, this term is exactly bounded by $V'$ (ignoring the $1 - 2\theta$ factor) and effectively sets the scale.

- $k = 2$: The mass is $\theta + \varepsilon$.

$$v_2 \log 3 = (\theta + \varepsilon)(1 - (\theta + \varepsilon)) \log 3$$
$$\leq (\theta + \varepsilon) \log 3.$$

We require $(\theta + \varepsilon) \log 3 \leq V' = 2\theta \log 2$. Since $\log 3 \approx 1.1$ and $2 \log 2 \approx 1.38$, strictly $\log 3 < 2 \log 2$. Thus, for sufficiently small $\varepsilon$ (specifically $\varepsilon \leq \theta(\frac{2 \log 2}{\log 3} - 1)$), this term is strictly less than $V'$.

- $k = 3$: The mass is $\theta - \varepsilon$.

$$v_3 \log 4 = (\theta - \varepsilon)(1 - (\theta - \varepsilon))2 \log 2 < 2\theta \log 2 = V'.$$

Thus, $V^*(p^{(j)}) \leq V'$ is satisfied for both hypotheses.

**Separation and KL Divergence.** The $\ell_\infty$ separation between the hypotheses is determined by the first two coordinates:

$$\|p^{(1)} - p^{(2)}\|_\infty = |(\theta + \varepsilon) - (\theta - \varepsilon)| = 2\varepsilon.$$

The KL divergence is bounded by the $\chi^2$ divergence:

$$D_{\mathrm{KL}}(p^{(1)} \| p^{(2)}) \leq \sum_{x \in \{1,2,3\}} \frac{(p_x^{(1)} - p_x^{(2)})^2}{p_x^{(2)}}$$
$$= \frac{(2\varepsilon)^2}{\theta - \varepsilon} + \frac{(-2\varepsilon)^2}{\theta + \varepsilon} + 0$$
$$= 4\varepsilon^2 \left( \frac{1}{\theta - \varepsilon} + \frac{1}{\theta + \varepsilon} \right) = \frac{8\varepsilon^2 \theta}{\theta^2 - \varepsilon^2}.$$

Assuming $\varepsilon \leq \theta/2$, we have $\theta^2 - \varepsilon^2 \geq \frac{3}{4}\theta^2$, so

$$D_{\mathrm{KL}}(p^{(1)} \| p^{(2)}) \leq \frac{8\varepsilon^2 \theta}{\frac{3}{4}\theta^2} = \frac{32\varepsilon^2}{3\theta}.$$

For the tensor product of $n$ samples, $D_{\mathrm{KL}}((p^{(1)})^{\otimes n} \| (p^{(2)})^{\otimes n}) = n D_{\mathrm{KL}}(p^{(1)} \| p^{(2)})$. We set this quantity to a constant, say $\frac{1}{2}$:

$$n \frac{32\varepsilon^2}{3\theta} \leq \frac{1}{2} \implies \varepsilon^2 \leq \frac{3\theta}{64n} \implies \varepsilon = c_1 \sqrt{\frac{\theta}{n}},$$

for some constant $c_1 = \frac{1}{8}\sqrt{\frac{3}{2}}$. Note that for large $n$, $\varepsilon \ll \theta$, satisfying our earlier small $\varepsilon$ assumptions.

**Lower Bound.** Applying Le Cam's inequality:

$$\inf_{\tilde{p}} \sup_{j \in \{1,2\}} \mathbb{E}_{p^{(j)}} \|\tilde{p} - p^{(j)}\|_\infty$$
$$\geq \frac{\|p^{(1)} - p^{(2)}\|_\infty}{4} \left( 1 - \sqrt{\frac{1}{2} D_{\mathrm{KL}}((p^{(1)})^{\otimes n} \| (p^{(2)})^{\otimes n})} \right)$$
$$\geq \frac{2\varepsilon}{4} \left( 1 - \frac{1}{2} \right) = \frac{\varepsilon}{4}.$$

Substituting $\varepsilon = c_1 \sqrt{\frac{\theta}{n}} = c_1 \sqrt{\frac{V'}{2n \log 2}}$:

$$\mathbb{E}_p \|\tilde{p} - p\|_\infty \geq \frac{c_1}{4} \sqrt{\frac{V'}{2n \log 2}} = c \sqrt{\frac{V'}{n}},$$

where $c = \frac{c_1}{4\sqrt{2 \log 2}}$ is a universal constant. This confirms the lower bound rate of $\sqrt{\frac{V'}{n}}$. $\qquad\square$

### 4.3. Proof of Lemma 2.5

*Proof.* **Lower bound:** Assume for contradiction that $M(p) < \frac{1}{3}$. Then for every $j$,

$$\frac{\log(j + 1)}{\log\left( 2 + \frac{\log(j+1)}{p_j n} \right)} < \frac{1}{3}$$
$$\implies \log\left( 2 + \frac{\log(j+1)}{p_j n} \right) > 3 \log(j + 1).$$

Exponentiating gives

$$2 + \frac{\log(j + 1)}{p_j n} > (j + 1)^3$$
$$\implies p_j < \frac{\log(j + 1)}{n\left( (j + 1)^3 - 2 \right)}.$$

Summing over $j$ yields

$$1 = \sum_{j \geq 1} p_j < \frac{1}{n} \sum_{j \geq 1} \frac{\log(j + 1)}{(j + 1)^3 - 2}.$$

Let $k = j + 1 \geq 2$. Then the sum is

$$\sum_{k \geq 2} \frac{\log k}{k^3 - 2} \leq \frac{4}{3} \sum_{k \geq 2} \frac{\log k}{k^3}$$
$$\leq \frac{4}{3} \left( \frac{\log 2}{2^3} + \int_2^\infty \frac{\log x}{x^3} \, dx \right)$$
$$= \frac{4}{3} \left( \frac{\log 2}{8} + \frac{\log 2}{8} + \frac{1}{16} \right) < 1.$$

Therefore $\sum_j p_j < \frac{1}{n} \cdot 1 \leq 1$, a contradiction. Hence $M(p) \geq \frac{1}{3}$.

**Upper bound:** Look at vector $p$ such that $p_i = \frac{1}{i}$ for all $i \in \mathbb{N}$. The functional $M(p)$ is defined for this vector in-spite of it not being a distribution.

We will first prove that for all $n \geq 2$, $M(p) < 5 \log n$.

Assume for contradiction that there exists $n' \geq 2$ such that for that $n$

$$M(p) > 5 \log n'. \tag{2}$$

Note that for all $n \geq 1$

$$\lim_{j \to \infty} \frac{\log(j+1)}{\log\left(2 + j\frac{\log(j+1)}{n}\right)} \leq \lim_{j \to \infty} \frac{\log(j+1)}{\log(1+j)} = 1.$$

Thus we can see that for all $n \geq 1$ the supremum in $M(p)$ is achieved.

For all $n \geq 1$ we can denote $f(n)$ as the $j$ value that is achieved by the supremum for that $n$. So for a given $n$

$$M(p) = \frac{\log(f(n)+1)}{\log\left(2 + \frac{f(n)\log(f(n)+1)}{n}\right)}.$$

We show that for previously mentioned $n'$, $f(n') > (n')^2$. $f(n') < (n')^2$ ,
Thus

$$M(p) \leq \frac{\log((n')^2+1)}{\log\left(2 + \frac{f(n')\log(f(n')+1)}{n'}\right)} \leq \frac{2\log(2n')}{\log(2)} =$$

$$= 2\frac{\log(n')}{\log(2)} + 1 \leq 5\log(n')$$

Where the last inequity is because $n' \geq 2$. This come in contradiction to 2.
Thus, $f(n') > (n')^2$. We can also see,

$$M(p) \leq \frac{\log(f(n')+1)}{\log(\frac{f(n')}{n'})} \leq \frac{\frac{3}{2}\log(f(n'))}{\log(\frac{f(n')}{n'})}$$

$$= \frac{3}{2}\left(1 + \frac{\log n'}{\log(f(n')) - \log(n')}\right)$$

$$\leq \frac{3}{2}\left(1 + \frac{\log n'}{2\log(n') - \log(n')}\right) = 3.$$

Where second transition is because $f(n') > 4$.
Next we see $5\log x > 3$ for $x \in [2,\infty)$. This is a contradiction to 2.

To complete the prof we will show that for all $p' \in \mathcal{P}(\mathbb{N})$ and all $n \geq 2$ $M(p) \geq M(p')$ and thus completing the proof.
Fix $p' \in \mathcal{P}(\mathbb{N})$ and $n \geq 2$. Without the loss of generality we treat $p$ as a sorted distribution. $\sum_j p'_j = 1$ and because it is a sorted distribution for all $j \in \mathbb{N}$, $p'_j \leq \frac{1}{j} = p'$.
In the case that the sup is achieved in $M(p')$ we denote $f(n)$ to be the function that gives us the $j$ that is chosen by the sup in $M(p')$. Thus,

$$M(p') = \frac{\log(f(n)+1)}{\log\left(2 + \frac{\log(f(n)+1)}{np'_{f(n)}}\right)}$$

$$\leq \frac{\log(f(n)+1)}{\log\left(2 + \frac{\log(f(n)+1)}{np_{f(n)}}\right)}$$

$$\leq \sup_j \frac{\log(j+1)}{\log\left(2 + \frac{\log(j+1)}{np_j}\right)} = M(p).$$

In the case that the sup not achieved in $M(p')$, using what we showed before,

$$M(p') = \lim_{j \to \infty} \frac{\log(j+1)}{\log\left(2 + \frac{\log(j+1)}{p'_j n}\right)} \tag{3}$$

$$\leq \lim_{j \to \infty} \frac{\log(j+1)}{\log\left(2 + j\frac{\log(j+1)}{n}\right)} \tag{4}$$

$$\leq 1 \leq 5\log n. \tag{5}$$

*Remark* 4.2. One can sharpen the constant: the best universal lower bound $\inf_{n \geq 1} \inf_p M(p)$ is $> 0$ and is achieved at $n = 1$; numerically it is about $0.48$. For $n$ large, $\inf_p M(p)$ increases and approaches 1 from below.

Now we show that the upper and lower bounds are achieved for some distributions. We look at $p_1$ such that $p_i = c\frac{1}{i^2}$ for all $i \geq 1$ and $c = \frac{6}{\pi^2}$.

Fix a sample size $n \geq 9$.
We look at $i = \sqrt{\frac{n}{\log n}}$

$$M(p) \geq \frac{\log(\sqrt{n/\log n} + 1)}{\log\left(2 + \frac{n}{\log n}\frac{\log(\sqrt{n/\log n}+1)}{cn}\right)}$$

$$\geq \frac{\log(\sqrt{n/\log n})}{\log\left(2 + \frac{\log(\sqrt{n/\log n}+1)}{c\log n}\right)}$$

Furthermore because it is trivial to prove that $\frac{\log(\sqrt{x/\log x}+1)}{\log x} \leq 1$ and $\frac{1}{2}\log x \leq \log x - \log\log x$ for all $x \geq \mathrm{e}^2$:

$$\frac{\log(\sqrt{n/\log n})}{\log\left(2 + \frac{\log(\sqrt{n/\log n}+1)}{c\log n}\right)} \geq$$

$$\frac{1}{2}\frac{\log(n) - \log\log n}{\log\left(2 + \frac{1}{c}\right)} \geq \frac{1}{2\log(2+1/c)} \cdot \frac{\mathrm{e}^2}{2}\log(n).$$

Now look at $p_2 = (\frac{1}{2}, \frac{1}{2}, 0\ldots)$

$$M(p) = \frac{\log(3)}{\log(2 + \frac{2\log(3)}{n})} \leq \frac{\log(3)}{\log(2)}.$$

$\square$

### 4.4. Proof of Lemma 2.6

*Proof.* **Upper bound.** Recall that $S(p) = \sup_{j \geq 1} p_j^\downarrow \log(j+1)$. Since $p^\downarrow$ is non-increasing and sums to 1, we have

$$1 = \sum_{k=1}^\infty p_k^\downarrow \geq \sum_{k=1}^j p_k^\downarrow \geq jp_j^\downarrow \implies p_j^\downarrow \leq \frac{1}{j}.$$

Thus,

$$p_j^\downarrow \log(j+1) \leq \frac{\log(j+1)}{j}.$$

The function $f(x) = \frac{\log(x+1)}{x}$ for $x \geq 1$ is maximized at $x \approx 2.16$ (since $f'(x) = \frac{x/(x+1) - \log(x+1)}{x^2}$, setting numerator to 0). For integer $j$, $f(1) = \log 2 \approx 0.69$, $f(2) = (\log 3)/2 \approx 0.55$. So $\sup_{j\geq 1} \frac{\log(j+1)}{j} \leq \log 2$. Hence $S(p) \lesssim 1$.

**Lower bound (unbounded near 0).** Consider the uniform distribution on $n$ elements, $u_n = (1/n, \dots, 1/n, 0, \dots)$. Then

$$S(u_n) = \sup_{1 \leq j \leq n} \frac{1}{n} \log(j+1) = \frac{\log(n+1)}{n}.$$

As $n \to \infty$, $S(u_n) \to 0$. Thus $S(p)$ can be arbitrarily small. $\square$

### 4.5. Proof of Theorem 2.4

*Proof.* The proof follows by refining the instance-dependent framework established in Blanchard & Voráček (2024), specifically their treatment of the local Glivenko-Cantelli (LGC) functionals. For a distribution $p \in \mathcal{P}(\mathbb{N})$, assume without loss of generality that $p = p^\downarrow$.

Let $X^n$ be a sample of $n$ i.i.d. observations from $p$, and let $\hat{p}$ be the empirical distribution. By permutation invariance of the sup-norm, we assume without loss of generality that $p = p^\downarrow$.

Define the one-sided maximum deviations $\Delta_n^+ := \sup_{i\geq 1}(\hat{p}_i - p_i)$ and $\Delta_n^- := \sup_{i\geq 1}(p_i - \hat{p}_i)$.

**Coordinate-wise deviation level.** Following Blanchard & Voráček (2024), define the base deviation level for coordinate $i$ as

$$\varepsilon_i := \inf\left\{\varepsilon \geq 0 : \mathbb{P}\left(\hat{p}_i \geq p_i + \varepsilon\right) \leq \frac{c_0}{2i}\right\},$$

where $c_0 \in (0, 1/4)$ is the universal constant from the anti-concentration bound in Lemma A.6. Let $\varepsilon := \sup_{i\geq 1} \varepsilon_i$. By definition, for every index $i$, the event $\hat{p}_i - p_i \geq \varepsilon$ occurs with probability at most $c_0/(2i)$.

**High-probability tail control.** We invoke the anti-concentration bound for $\beta = 2$ (Lemma A.6) which states that for $\hat{p}_i \sim n^{-1}\mathrm{Bin}(n, p_i)$, there exists $C' \geq 1$ such that

$$\mathbb{P}(\hat{p}_i \geq p_i + \varepsilon) \geq c_0 e^{-C'nD_{\mathrm{KL}}(p_i+\varepsilon||p_i)}.$$

Combining this with the definition of $\varepsilon_i$, we have for the adjusted term $\tilde{\varepsilon} := (\varepsilon \wedge 1/4) \vee (1/n)$:

$$\mathbb{P}(\hat{p}_i \geq p_i + \varepsilon) \geq \mathbb{P}(\hat{p}_i \geq p_i + \tilde{\varepsilon}) \geq c_0 e^{-C'nD_{\mathrm{KL}}(p_i+\tilde{\varepsilon}||p_i)}.$$

$$D_{\mathrm{KL}}\left(p_i + \tilde{\varepsilon}||p_i\right) \geq \frac{\log(2i)}{nC'}.$$

By the convexity of the Kullback-Leibler divergence $q \mapsto D_{\mathrm{KL}}(q||p)$, for any scaling factor $k \geq 1$, it holds that $D_{\mathrm{KL}}(p_i + kC'\tilde{\varepsilon}||p_i) \geq kC'D_{\mathrm{KL}}(p_i + \tilde{\varepsilon}||p_i) \geq k\frac{\log(2i)}{n}$. Applying the Chernoff bound (Lemma A.5) yields

$$\mathbb{P}(\hat{p}_i - p_i \geq kC'\tilde{\varepsilon}) \leq e^{-nD_{\mathrm{KL}}\left(p_i+kC'\tilde{\varepsilon}||p_i\right)} \leq \frac{1}{(2i)^k}. \quad (6)$$

**Union bound and summability.** Fix $\delta \in (0, e^{-2}]$ and let $k^{(\delta)} = \frac{\log(2/\delta)}{\log 2}$. Note that $\delta \leq e^{-2}$ implies $k^{(\delta)} \geq 2$. Applying the union bound over all coordinates $i \geq 1$:

$$\mathbb{P}(\Delta_n^+ \geq k^{(\delta)}C'\tilde{\varepsilon}) \leq \sum_{i=1}^\infty \mathbb{P}(\hat{p}_i - p_i \geq k^{(\delta)}C'\tilde{\varepsilon})$$

$$\overset{(6)}{\leq} \sum_{i=1}^\infty \frac{1}{(2i)^{k^{(\delta)}}} = \frac{1}{2^{k^{(\delta)}}} \sum_{i=1}^\infty \frac{1}{i^{k^{(\delta)}}}.$$

For $k^{(\delta)} \geq 2$, the sum is bounded by $1 + \int_1^\infty x^{-2}dx = 2$. Thus,

$$\mathbb{P}(\Delta_n^+ \geq k^{(\delta)}C'\tilde{\varepsilon}) \leq \frac{2}{2^{k^{(\delta)}}} = \frac{2}{2^{\log(2/\delta)/\log 2}} = \delta.$$

This establishes that $\Delta_n^+ \leq C\varepsilon \log(1/\delta)$ with probability at least $1 - \delta$.

We first note a simple symmetry that allows us to restrict attention to the case $p_i \leq 1/2$. For each coordinate,

$$|\hat{p}_i - p_i| \overset{d}{=} \left|n^{-1}\mathrm{Bin}(n, p_i) - p_i\right|,$$

and this distribution is unchanged when $p_i$ is replaced by $1 - p_i$. Hence the coordinatewise tail probabilities appearing in the union bound are unchanged if, in that step, we replace $p$ by the vector $p'$ defined by

$$p_i' = \begin{cases} 1 - p_i, & p_i > 1/2, \\ p_i, & p_i \leq 1/2. \end{cases}$$

Moreover, $p_i' \leq p_i$ for every $i$, and therefore also $(p')_i^\downarrow \leq p_i^\downarrow$ for all $i$. Consequently,

$$S(p') \leq S(p), \qquad M(p') \leq M(p).$$

Thus, proving the desired union-bound estimate under the assumption $p_i \in [0, 1/2]$ for all $i$ also proves it for the original vector $p$.

**Symmetry and left-tail behavior.** Since $p_i \leq 1/2$, the binomial distribution $\mathrm{Bin}(n, p_i)$ has a heavier right tail than left tail. Formally, for any $\varepsilon > 0$, $D_{\mathrm{KL}}(p_i - \varepsilon||p_i) \geq D_{\mathrm{KL}}(p_i + \varepsilon||p_i)$ (Garivier & Cappé, 2011). Consequently, the same union bound argument applies to $\Delta_n^-$, yielding $\mathbb{P}(\Delta_n^- \geq C\varepsilon \log(1/\delta)) \leq \delta$. By the union bound over the left and right events, $\|\hat{p} - p\|_\infty \leq C\varepsilon \log(1/\delta)$ with probability at least $1 - 2\delta$.

**Characterization of the deviation level.** It remains to relate $\varepsilon = \sup_i \varepsilon_i$ to the functionals $S(p)$ and $M(p)$. By Proposition 13 in Blanchard & Voráček (2024), the deviation level satisfies $\varepsilon_i \leq C_1 \phi_{i,p_i}(n)$. From the definition of $\phi$, the supremum over $i$ is bounded by the sub-Gaussian and sub-gamma terms, yielding

$$\varepsilon \lesssim \left( \sqrt{\frac{S(p)}{n}} \vee \frac{M(p)}{n} \right) \wedge 1.$$

Recall that $\tilde{\varepsilon} = (\varepsilon \wedge 1/4) \vee (1/n)$. By Lemma 2.5, $M(p) \gtrsim 1$, which implies $\varepsilon \gtrsim 1/n$. Since we also have $\varepsilon \lesssim 1$ in the range of interest, it follows that $\tilde{\varepsilon} \approx \varepsilon$, establishing the final bound. $\square$

### 4.5.1. PROOF OF LEMMA 4.3

**Lemma 4.3** (M-term can dominate S-term). *There exist distributions $p$ for which $M(p)$ strictly dominates $\sqrt{S(p)}$ for large $n$.*

*Proof of Lemma 4.3.* Fix a sample size $n \geq 1$. Consider the uniform distribution $p$ such that $p_j = \frac{1}{n}$ for $j \leq n$.

$$\frac{M(p)}{n} = \frac{\log(n+1)}{n} \cdot \left( \log \left( 2 + \frac{n \log(n+1)}{n} \right) \right)^{-1}$$

The index $n$ is chosen because $\frac{\log x}{\log(2+\log x)}$ is an increasing function. Meanwhile, the $S$-term is

$$\sqrt{\frac{S(p)}{n}} = \sqrt{\frac{\log n}{n^2}}.$$

Since $M(p)/n$ is larger than $\sqrt{S(p)/n}$ for large $n$, with $M(p) > \sqrt{S(p)}$ (in term comparison) at $n = 1$, the result follows. $\square$

## Impact Statement

This paper presents work whose goal is to advance the field of Machine Learning. There are many potential societal consequences of our work, none which we feel must be specifically highlighted here.

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

# A. Additional Results & Auxiliary Lemmas

**Lemma A.1** (Negative covariance of multinomial components; Dubhashi & Ranjan (1998)). *Let $\{X^{(j)}\}_{j=1}^n$ be $n$ i.i.d. samples from a multinomial distribution with parameter vector $\mathbf{p} = (p_1, p_2, \dots)$ and single trial (i.e., $X^{(j)} \sim \text{Multinomial}(1, \mathbf{p})$). Let $X_{i,j}$ denote the $i$-th component of the vector $X^{(j)}$. Define*

$$Z_i = \frac{1}{n}\sum_{j=1}^n X_{i,j} - p_i, \qquad Z_i' = p_i - \frac{1}{n}\sum_{j=1}^n X_{i,j}.$$

*Then for all distinct $i \neq k$ with $p_i, p_k > 0$,*

$$\text{Cov}(Z_i, Z_k) = -\frac{p_i p_k}{n} < 0 \quad and \quad \text{Cov}(Z_i', Z_k') < 0. \tag{7}$$

*Moreover, the variables $\{Z_i\}_i$ are Negatively Associated in the sense of Dubhashi & Ranjan (1998).*

## A.1. Proof of Theorem 2.2

This lemma establishes a continuity-like property for V*, which is crucial for relating the population functional to its empirical version in the high-probability bound.

### A.1.1. PROOF OF LEMMA A.2

**Lemma A.2** (Distribution-dependent bound on $V^*$). *For two distributions $p, q$, we have*

$$\Delta(p, q) \leq \bar{\delta}(p, q) \log \frac{1}{\bar{\delta}(p, q)},$$

*where $\Delta(p, q) := |V^*(p) - V^*(q)|$ and $\bar{\delta}(p, q) := \sup_{i \in \mathbb{N}} |v_i(p^\downarrow) - v_i(q^\downarrow)|$. In particular, since $\bar{\delta}(p, q) \leq \|p - q\|_\infty$, we have $\Delta(p, q) \leq \|p - q\|_\infty \log(1/\|p - q\|_\infty)$.*

*Proof.* Recall that $V^*(p) = \sup_{i \geq 1} v_i(p^\downarrow) \log(i + 1)$. Using the inequality $|\sup_i a_i - \sup_i b_i| \leq \sup_i |a_i - b_i|$, we have

$$\Delta(p, q) \leq \sup_{i \geq 1} \left| v_i(p^\downarrow) \log(i + 1) - v_i(q^\downarrow) \log(i + 1) \right|.$$

Let $\delta_i = |v_i(p^\downarrow) - v_i(q^\downarrow)|$. From Kontorovich & Painsky (2025, Lemma 12), we know that $v_i(p^\downarrow) \leq \frac{1}{i+1}$ for any distribution $p$. Since $x, y \geq 0 \implies |x - y| \leq \max(x, y)$, we have

$$\delta_i \leq \max(v_i(p^\downarrow), v_i(q^\downarrow)) \leq \frac{1}{i + 1},$$

which implies $\log(i+1) \leq \log(1/\delta_i)$. The function $f(x) = x \log(1/x)$ is increasing on $(0, 1/e]$. Since $\delta_i \leq 1/2$, we have

$$\delta_i \log(i + 1) \leq \delta_i \log \frac{1}{\delta_i} \leq \bar{\delta}(p, q) \log \frac{1}{\bar{\delta}(p, q)}.$$

Taking the supremum over $i$ yields the first claim. For the second, note that $|v_i(p^\downarrow) - v_i(q^\downarrow)| \leq |p_i^\downarrow - q_i^\downarrow| \leq \|p^\downarrow - q^\downarrow\|_\infty \leq \|p - q\|_\infty$. $\qquad\square$

### A.1.2. PROOF OF LEMMA A.3

**Lemma A.3** (Empirical functional concentration). *With probability $\geq 1 - \delta - \frac{81}{n}$, we have*

$$\delta(p, \hat{p}) \leq a + \frac{3b^2}{2} + b\sqrt{a} + \frac{3b\sqrt{\hat{v}^*}}{2},$$

*where*

$$a = \frac{4}{3n} \log \frac{2(n + 1)}{\delta} + \frac{\log n}{n},$$
$$b = 2\sqrt{\frac{\log(n + 1)}{n} + \frac{1}{n} \log \frac{2}{\delta}}.$$

*Proof.* We will show that

$$\delta(p, \hat{p}) \leq \|p - \hat{p}\|_\infty \leq a + b\sqrt{\hat{v}^*} + b\sqrt{\delta(p, \hat{p})}$$

and from there apply the proof technique from Kontorovich & Painsky (2025, Theorem 4). First, it follows from that proof that

$$\|p - \hat{p}\|_\infty \leq a + b\sqrt{\hat{v}^*} + b\sqrt{|v^* - \hat{v}^*|}$$
$$\leq a + b\sqrt{\hat{v}^*} + b\sqrt{\delta(p, \hat{p})}.$$

Next, for all distributions $p, q$ we have

$$|p_i(1 - p_i) - q_i(1 - q_i)|$$
$$= |p_i - q_i + (q_i - p_i)(q_i + p_i)|$$
$$\leq |p_i - q_i|.$$

Going back to $\delta(p, \hat{p})$, where $i$ is the index chosen by the supremum (note that sup and max are equivalent in this case)

$$\delta(p, \hat{p}) = \sup_{i \in \mathbb{N}} |p_i(1 - p_i) - q_i(1 - q_i)|$$
$$\leq |p_i - q_i| \leq \|p - \hat{p}\|_\infty.$$

Now we have:
$$A \leq B\sqrt{A} + C,$$

where $A = \delta(p, \hat{p})$, $B = b$, $C = a + b\sqrt{\hat{v}^*}$, which implies $A \leq B^2 + B\sqrt{C} + C$, or

$$\delta(p, \hat{p}) \leq b^2 + a + b\sqrt{\hat{v}^*} + b\sqrt{a + b\sqrt{\hat{v}^*}}.$$

Using $\sqrt{x + y} \leq \sqrt{x} + \sqrt{y}$ and $\sqrt{xy} \leq (x + y)/2$,

$$\delta(p, \hat{p}) \leq b^2 + a + b\sqrt{\hat{v}^*} + b\sqrt{a} + b\sqrt{b\sqrt{\hat{v}^*}}$$

$$\leq b^2 + a + b\sqrt{\hat{v}^*} + b\sqrt{a} + \frac{b(b + \sqrt{\hat{v}^*})}{2}$$

$$= a + \frac{3b^2}{2} + b\sqrt{a} + \frac{3b\sqrt{\hat{v}^*}}{2}.$$

We still have

$$a + b\sqrt{v^*} \leq a + \frac{3b^2}{2} + b\sqrt{a} + \frac{3b\sqrt{\hat{v}^*}}{2}.$$

whence, with probability $1 - \delta - \frac{81}{n}$,

$$\|p - \hat{p}\|_\infty \leq a + \frac{3b^2}{2} + b\sqrt{a} + \frac{3b\sqrt{\hat{v}^*}}{2}.$$

$\square$

**Lemma A.4** (Global Scaling of Bernoulli KL Divergence).
*Fix $p \in (0, 1/2]$. For any $x > 0$ and $y \geq 1$ satisfying $p + xy \leq 1/2$, the following inequality holds:*

$$D_{\mathrm{KL}}\left(p + xy\|p\right) \leq 8y^2 D_{\mathrm{KL}}\left(p + x\|p\right).$$

*Proof.* Let $u := x/p$. We seek to bound the ratio of divergences. We first establish global bounds for the numerator and denominator.

**Step 1: Global lower bound for the denominator.** Let $t > 0$ such that $p + t \leq 1/2$. We claim that

$$D_{\mathrm{KL}}\left(p + t\|p\right) \geq \frac{t}{2} \log\left(1 + \frac{t}{p}\right). \tag{8}$$

*Proof of* (8)*:* Using the inequality $\log(1 - v) \geq -v/(1 - v)$ for $v < 1$, we have

$$D_{\mathrm{KL}}\left(p + t\|p\right) = (p + t)\log(1 + t/p)$$
$$+ (1 - p - t)\log\left(1 - \frac{t}{1 - p}\right)$$
$$\geq (p + t)\log(1 + t/p) - t.$$

Let $z = t/p$. The RHS is $p[(1 + z)\log(1 + z) - z]$. Consider the function

$$g(z) = (1 + z)\log(1 + z) - z - \frac{z}{2}\log(1 + z)$$
$$= \left(1 + \frac{z}{2}\right)\log(1 + z) - z.$$

Differentiation yields

$$g'(z) = \frac{1}{2}\log(1 + z) + \frac{1 + z/2}{1 + z} - 1$$
$$= \frac{1}{2}\log(1 + z) - \frac{z}{2(1 + z)}.$$

Since $\log(1 + z) \geq \frac{z}{1+z}$, we have $g'(z) \geq 0$. Since $g(0) = 0$, $g(z) \geq 0$ for all $z \geq 0$. Thus, $D_{\mathrm{KL}}\left(p + t\|p\right) \geq p \cdot \frac{z}{2}\log(1 + z) = \frac{t}{2}\log(1 + t/p)$.

**Step 2: Case analysis.** We split the analysis based on the magnitude of the shift $xy$ relative to $p$.

**Case 1:** $xy \leq p$. This implies $x \leq p$ (since $y \geq 1$). For the numerator, we use the quadratic upper bound derived from Taylor expansion. Since $p \leq 1/2$, we have $1 - p \geq 1/2$. Using $\log(1 + a) \leq a$ and $\log(1 - b) \leq -b$:

$$D_{\mathrm{KL}}\left(p + xy\|p\right) \leq \frac{(xy)^2}{p} + \frac{(xy)^2}{1 - p}$$
$$\leq \frac{(xy)^2}{p} + \frac{2(xy)^2}{p} \leq \frac{2(xy)^2}{p}.$$

For the denominator, since $x \leq p$, we have $u = x/p \leq 1$. Using $\log(1 + u) \geq u/2$ for $u \in [0, 1]$ in (8):

$$D_{\mathrm{KL}}\left(p + x\|p\right) \geq \frac{x}{2} \cdot \frac{u}{2} = \frac{x^2}{4p}.$$

The ratio is bounded by:

$$\frac{D_{\mathrm{KL}}\left(p + xy\|p\right)}{D_{\mathrm{KL}}\left(p + x\|p\right)} \leq \frac{2x^2 y^2/p}{x^2/4p} = 8y^2.$$

**Case 2:** $xy > p$. In this regime, we use the bound $D_{\mathrm{KL}}\left(p + t\|p\right) \leq (p + t)\log(1 + t/p)$. Thus,

$$D_{\mathrm{KL}}\left(p + xy\|p\right) \leq (p + xy)\log(1 + xy/p).$$

We distinguish two sub-cases based on $x$.

*Sub-case 2a: $x > p$.* Here, both numerator and denominator are in the "linear-log" regime. From (8), $D_{\mathrm{KL}}\left(p + x\|p\right) \geq \frac{x}{2}\log(1 + x/p)$. Let $u = x/p > 1$.

$$\frac{D_{\mathrm{KL}}\left(p + xy\|p\right)}{D_{\mathrm{KL}}\left(p + x\|p\right)} \leq \frac{p(1 + uy)\log(1 + uy)}{\frac{pu}{2}\log(1 + u)}.$$

Since $u > 1$ and $y \geq 1$, we have $\frac{1+uy}{u} \leq 2y$ and $\frac{\log(1+uy)}{\log(1+u)} \leq 1 + \frac{\log y}{\log(1+u)} \leq 1 + 1.5\log y$. Thus the ratio is at most $2(2y)(1 + 1.5\log y) = 4y + 6y\log y$. Since $y \geq 1$, we verify that $4y + 6y\log y \leq 8y^2$. (At $y = 1$, $4 \leq 8$; for $y > 1$, $y^2$ grows faster than $y\log y$.)

*Sub-case 2b: $x \leq p$.* Here $x$ is small but $xy$ is large (transition regime). Numerator: $D_{\mathrm{KL}}\left(p + xy\|p\right) \leq (p + xy)\log(1 + xy/p) \leq 2xy\log(1 + xy/p)$ (since $p < xy$). Denominator: $D_{\mathrm{KL}}\left(p + x\|p\right) \geq \frac{x^2}{4p}$ (as derived in Case 1).

$$\text{Ratio} \leq \frac{2xy\log(1 + xy/p)}{x^2/4p} = \frac{8py}{x}\log\left(1 + \frac{xy}{p}\right).$$

Let $u = x/p \leq 1$. Then Ratio $\leq \frac{8y}{u}\log(1 + yu)$. Since $\log(1 + z) \leq z$ for all $z$,

$$\text{Ratio} \leq \frac{8y}{u}(yu) = 8y^2.$$

**Conclusion.** In all cases, the ratio is bounded by $8y^2$. □

**Lemma A.5** (Chernoff bound for Binomial variables). *Let $n \in \mathbb{N}$ and let $Y \sim \mathrm{Bin}(n,p)$ with $p \in [0,1]$. Then for every $q \in [p,1]$,*

$$\Pr\left(\frac{Y}{n} \geq q\right) \leq \exp(-n\,D_{\mathrm{KL}}\,(q||p)),$$

*where for $p,q \in [0,1]$,*

$$D_{\mathrm{KL}}\,(q||p) := q\log\frac{q}{p} + (1-q)\log\frac{1-q}{1-p}.$$

**Lemma A.6** (Anti-concentration for Binomial upper tails (Zhang & Zhou, 2020)). *For any $\beta > 1$ there exist constants $c_\beta, C_\beta > 0$, depending only on $\beta$, such that*

$$\mathbb{P}(\hat{p}_n - p \geq \varepsilon) \geq c_\beta \exp\left(-C_\beta n D_{\mathrm{KL}}\,(\varepsilon + p||p)\right),$$
$$\text{if } 0 \leq \varepsilon \leq \frac{1-p}{\beta} \text{ and } \varepsilon + p \geq \frac{1}{n};$$
$$= 1 - (1-p)^n, \quad \text{if } 0 < \varepsilon + p < \frac{1}{n}.$$

**Lemma A.7** (Blanchard & Voráček (2024)). *Let $0 \leq q, \varepsilon \leq \frac{1}{4}$ and suppose $\varepsilon \geq 8q$. Define $h(u) := (1+u)\ln(1+u) - u$. Then,*

$$\frac{\varepsilon}{2}\ln\frac{\varepsilon}{q} \leq qh\left(\frac{\varepsilon}{q}\right) \leq D_{\mathrm{KL}}\,(q+\varepsilon||q) \leq 2\varepsilon\ln\frac{\varepsilon}{q}.$$

*Also, for any $0 \leq q, \varepsilon \leq 1$ with $q + \varepsilon \leq \frac{1}{2}$,*

$$\frac{\varepsilon^2}{2(q+\varepsilon)} \leq qh\left(\frac{\varepsilon}{q}\right) \leq D_{\mathrm{KL}}\,(q+\varepsilon||q) \leq \frac{\varepsilon^2}{q}.$$

### A.2. Anti-concentration Lower Bound

**Proposition A.8.** *Let $S(p) := \sup_{j \geq 1} p_j \log(j+1)$ and let $j'$ be an index such that $p_{j'}\log(j'+1) \geq S(p)/2$. Then for any $k > 0$, we have*

$$\mathbb{P}\left(\sup_j(\hat{p}_j - p_j) \geq \tau\right) \geq (1-e^{-1})\left(1 \wedge \frac{c_2}{2(j'+1)^{2k^2-1}}\right),$$

*where $\tau := \left(k\sqrt{\frac{S(p)}{2Cn}} \wedge \frac{1}{4}\right) \vee \frac{1}{n}$, and $c_2, C$ are universal constants from Lemma A.6.*

*Proof.* Let $\tau := \left(k\sqrt{\frac{S(p)}{2Cn}} \wedge \frac{1}{4}\right) \vee \frac{1}{n}$. Note that for all $j$, $p_j + \tau \geq \frac{1}{n}$. Invoking Lemma A.7 and Lemma A.6, we have that

$$\mathbb{P}(\hat{p}_j - p_j \geq \tau) \geq c_2 \exp(-C_2 n\, D_{\mathrm{KL}}\,(p_j + \tau||p_j)).$$

In the regime where $\tau = k\sqrt{\frac{S(p)}{2Cn}}$, using Lemma A.4 yields

$$\mathbb{P}(\hat{p}_j - p_j \geq \tau) \geq c_2 \exp\left(-C_2 n\frac{k^2 S(p)}{2Cnp_j}\right)$$
$$= c_2 \exp\left(-k^2\frac{S(p)}{p_j}\right).$$

As discussed in Theorem 2.1, the estimators $\hat{p}_j$ are negatively associated. This implies that for the increasing events $A_j = \{\hat{p}_j - p_j \geq \tau\}$, we have $\mathbb{P}(\cap A_j^c) \leq \prod \mathbb{P}(A_j^c)$. Consequently,

$$\mathbb{P}(\cup A_k) \geq (1-e^{-1})(1 \wedge \sum \mathbb{P}(A_k)):$$

$$\mathbb{P}\left(\sup_j(\hat{p}_j - p_j) \geq \tau\right) \geq (1-e^{-1})\left(1 \wedge \sum_j c_2 \times\right.$$
$$\left.\exp\left(-k^2\frac{S(p)}{p_j}\right)\right).$$

By definition of $S(p) = \sup_{j \geq 1} p_j \log(j+1)$, there exists an index $j'$ such that

$$p_{j'}\log(j'+1) \geq \frac{1}{2}S(p) \implies \frac{S(p)}{p_{j'}} \leq 2\log(j'+1).$$

Since $p$ is sorted non-increasingly, for all $j \leq j'$, $p_j \geq p_{j'}$. Thus, we can lower bound the sum by truncating at $j'$:

$$\sum_j c_2 \exp\left(-k^2\frac{S(p)}{p_j}\right) \geq \sum_{j=1}^{j'} c_2 \exp\left(-k^2\frac{S(p)}{p_{j'}}\right)$$
$$\geq j'c_2 \exp\left(-2k^2\log(j'+1)\right)$$
$$= \frac{c_2 j'}{(j'+1)^{2k^2}}$$
$$\geq \frac{c_2}{2(j'+1)^{2k^2-1}}.$$

Substituting this back into our lower bound for the probability of the union:

$$\mathbb{P}\left(\sup_j(\hat{p}_j - p_j) \geq \tau\right) \geq (1-e^{-1}) \times$$
$$\left(1 \wedge \frac{c_2}{2(j'+1)^{2k^2-1}}\right).$$

This provides a polynomial lower bound in terms of the truncation index $j'$, which is related to the tail decay of the distribution. □

## B. Experiments

We compare our fully empirical high-probability bound from Theorem 2.2 to the bounds of Kontorovich & Painsky (2025), reproducing their experimental protocol and figures, and then adding our bound as an additional curve.

**Protocol (matching Kontorovich & Painsky (2025)).** For each sample size $n$, we draw $R = 1000$ i.i.d. samples $X^n \sim p$, compute the empirical pmf $\hat{p}$, and evaluate each candidate upper bound on $\|\hat{p} - p\|_\infty$. We also plot an *oracle reference curve* defined as the empirical $(1 - \delta)$-quantile of $\|\hat{p} - p\|_\infty$ across the $R$ repetitions (this is not a bound computable from data, but serves as a scale reference). All curves are shown on a log–log scale in $n$.

**Distributions and datasets.** We use the same synthetic distributions as Kontorovich & Painsky (2025): (i) the uniform distribution over $A = 100$ symbols, and (ii) a truncated Zipf law with parameter $s = 1.1$ over $A = 100$ symbols. We also reproduce the two real-world datasets from Kontorovich & Painsky (2025) (Surnames and Hamilton), treating the dataset's empirical histogram as the ground-truth distribution $p$ and repeatedly resampling $X^n$ from this $p$.

**Confidence levels and sample-size grids.** We consider two choices of confidence: (a) fixed $\delta = 0.05$ (Figure 1a and Figure 1c), and (b) $\delta = 1/n^2$ (Figure 1b). The grids of $n$ match the reproduction notebook: Figure 1a uses $n \in \{10^4, 2 \cdot 10^4, \ldots, 2 \cdot 10^5\}$, Figure 1b uses $n \in \{10^5, 2 \cdot 10^5, \ldots, 10^6\}$, and Figure 1c uses $n \in \{10^4, 2 \cdot 10^4, \ldots, 5 \cdot 10^5\}$.

**Bounds compared.** We plot the two bounds from Kontorovich & Painsky (2025) that appear in their figures (labeled "K–P (Thm 2)" and "K–P (Thm 4)"), their benchmark curve, and our new bound. Our curve ("Ours (Thm 2.2)") is obtained by evaluating the fully empirical plug-in quantities $\hat{V}^*$ and $\hat{v}^*$ from Section 2, as in Theorem 2.2. For the plots we instantiate Theorem 2.2 using the explicit constants currently implemented in the code (a direct transcription of the current proof-level constants):

$$\mathrm{Bound}_{\mathrm{ours}}(\hat{p}, n, \delta)$$

$$= 2\sqrt{\frac{\hat{V}^*}{n} + \frac{\hat{v}^*}{n}\log\frac{2}{\delta}} + \frac{1}{n}\log\left(\frac{6n(n+1)}{\delta}\right)$$

$$+ \frac{4}{3n}\log\left(\frac{2(n+1)}{\delta}\right) + \frac{\log n}{n}.$$

(These constants are not optimized; the purpose of the experiments is a like-for-like comparison under the same plotting protocol.)

**Results.** Our fully empirical bound consistently improves upon the benchmark curve across all distributions and confidence levels. In settings with heavier tails (Zipf and real-world datasets), our bound substantially outperforms the competing empirical bound "K–P (Thm 4)" and approaches the tightness of the population-dependent "K–P (Thm 2)" bound (Figure 1a (right) and Figure 1c). For the uniform distribution, where the tail complexity is minimal, "K–P (Thm

4)" remains tighter than our bound (Figure 1a (left)). In the vanishing-confidence regime $\delta = 1/n^2$, our bound is robust, particularly for the Zipf distribution where it nearly matches the population-based "K–P (Thm 2)" curve (Figure 1b).

Figure 1: $A = 100$, $\delta = 0.05$ (log-log scale)

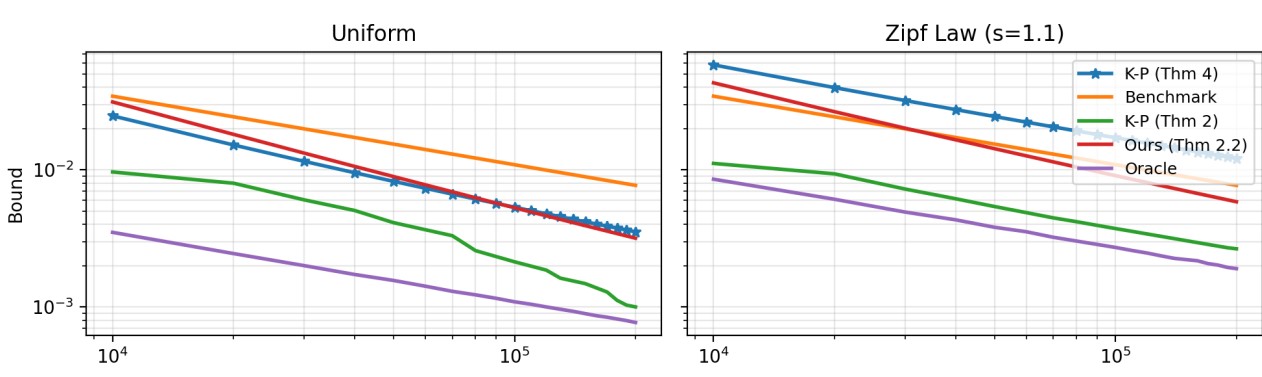

*(a)* Synthetic distributions with $A = 100$ and fixed confidence $\delta = 0.05$ (log–log scale).

Figure 2: $A = 100$, $\delta = 1/n^2$ (log-log scale)

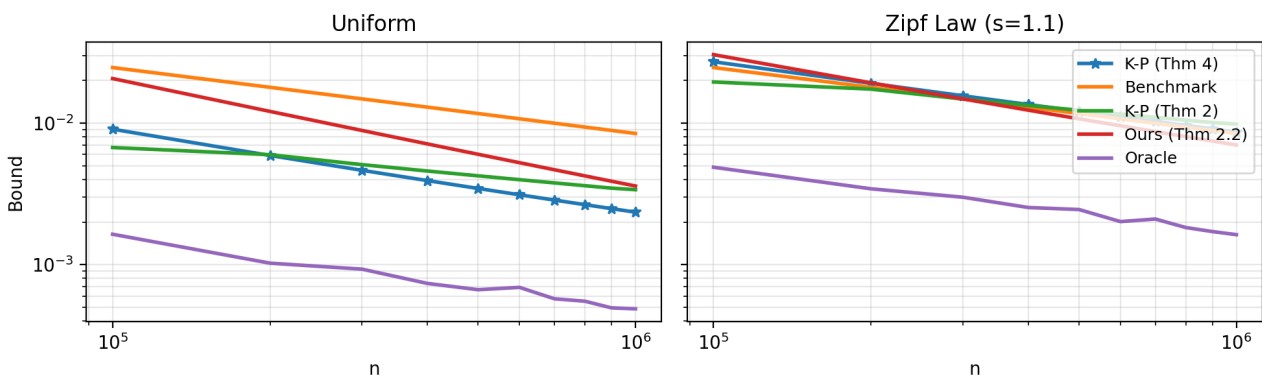

*(b)* Synthetic distributions with $A = 100$ and $\delta = 1/n^2$ (log–log scale).

Figure 3: real-world datasets, $\delta = 0.05$ (log-log scale)

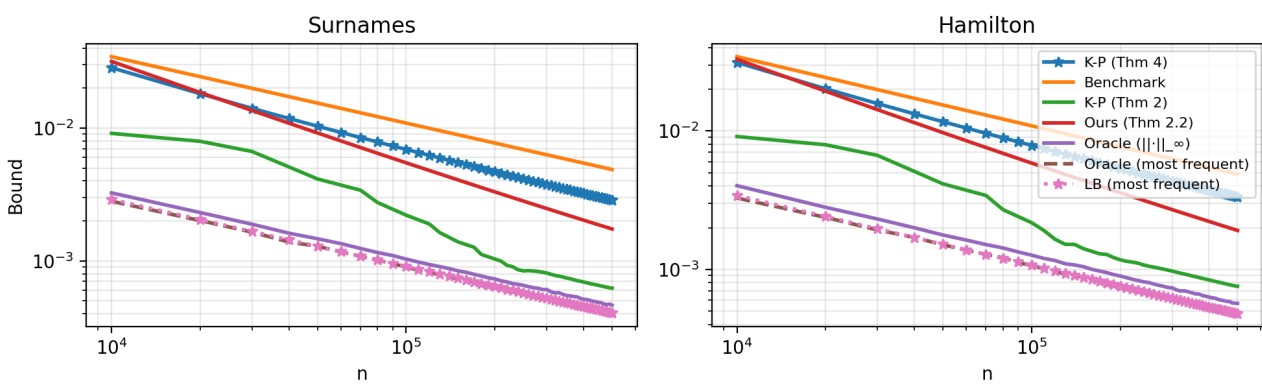

*(c)* Real-world datasets (Surnames and Hamilton) with $\delta = 0.05$ (log–log scale).

*Figure 1.* Comparison of bounds reproducing Kontorovich & Painsky (2025) and adding our bound from Theorem 2.2.

