# OpenReview forum: "Improved Distribution Estimation in $\ell_\infty$"
_ICML.cc/2026/Conference — ICML 2026 regular_

### Official Review · Reviewer_uGKo · 2026-03-07

**Soundness:** 2
**Presentation:** 1
**Significance:** 2
**Originality:** 2
**Overall Recommendation:** 3
**Confidence:** 4

**Summary:**

This paper investigates the $\ell_{\infty}$ error of distribution estimation over a countable alphabet. The paper prove that the worst-case expected risk is determined by a two-point distribution $p^* = (1/2, 1/2, 0, 0, \ldots)$, thereby establishing the global minimax rate of $\Theta(n^{-1/2})$. The paper further proposes a fully empirical high-probability bound, which replaces unknown population functionals with observable statistics $\hat{V}^* $ and $\hat{v}^* $. Based on local Glivenko–Cantelli theory, the study also derives distribution-dependent upper bounds, revealing that the estimation error consists of a two-part structure: a variance term $\sqrt{S(p)/n}$ and a tail complexity term $M(p)/n$.

**Compliance With Llm Reviewing Policy:**

Affirmed.

**Final Justification:**

The rebuttal clarified several important points and increased my confidence in the paper’s technical core. While I still have reservations about the presentation, limited empirical validation, and broader significance, these concerns remain strong enough that I retain a weak reject recommendation.

**Key Questions For Authors:**

* The paper’s motivation and problem formulation are highly related to the work of Kontorovich & Painsky (2025). Could the authors further clarify what the research value of this work is, beyond complementing or improving upon that prior work,  lies in broader theoretical significance or potential practical value?
* In the introduction section, the paper cites only one related work. Is this sufficient to adequately present the background of the problem? Should additional references be included to more fully elaborate on the problem background, its significance, and the current theoretical developments in this area?
* Several theoretical results depend on distribution-dependent quantities related to the tail behavior (e.g., $(S(p))$, $(M(p))$). Could the authors provide more intuition or discussion on how these quantities behave for common distributions and whether they can be reliably estimated in practice?

**Limitations:**

Yes. The paper notes that its analysis is developed for discrete distributions over countable alphabets and focuses primarily on theoretical guarantees for $\ell_{\infty}$ distribution estimation. In addition, the results also rely on distribution-dependent quantities such as $(S(p))$ and $(M(p))$, whose behavior and estimability in practical settings are not fully explored. In addition, the empirical evaluation is relatively limited, leaving the practical usefulness of the proposed bounds less clearly demonstrated.

**Strengths And Weaknesses:**

**Strengths:**
* **Minimax characterization:** The paper establishes the minimax convergence rate $\Theta(n^{-1/2})$ for $\ell_{\infty}$ distribution estimation and identifies a simple two-point distribution as least-favorable.
* **Fully empirical high-probability bound:** The proposed deviation bound depends only on observable plug-in statistics, making the guarantee data-dependent and directly computable from the samples.
* **Refined distribution-dependent insight:** By leveraging local Glivenko–Cantelli theory, the paper reveals that the estimation error is jointly governed by a variance term and a tail-complexity term, providing a clearer picture of the difficulty across different tail regimes.

**Weakness:**
* **Soundness:** Several key proof steps rely on reductions to existing results and auxiliary lemmas, with limited explanation in the main text . In addition, the empirical evaluation is relatively limited and mainly illustrative, leaving the tightness and practical behavior of the bounds insufficiently validated.
* **Presentation:** The paper places experimental results in the appendix while substantial technical proofs remain in the main text, making it harder for readers to quickly grasp the main contributions. In addition, the logical connections among the theoretical results are not always clearly articulated, giving the impression of a collection of separate results rather than a coherent narrative. The related work discussion is also somewhat limited (e.g., referring readers to “the works cited therein”), and the manuscript contains several typographical errors (e.g., “tail/complexity,” “0there”).
* **Significance:** While the paper provides refined theoretical analysis for the $\ell_{\infty}$ distribution estimation problem, the broader impact of these results is somewhat unclear. The work mainly improves theoretical bounds, and the paper does not clearly demonstrate how these results would translate into practical improvements for real-world estimation tasks or influence downstream applications.
* **Originality:** The paper builds heavily on the framework and results of the prior work of Kontorovich and Painsky, and many components of the paper can be viewed as refinements or extensions of existing analyses. As a result,  the overall level of conceptual novelty appears somewhat limited.

---

> ### Author Rebuttal · Authors · 2026-03-30
>
> We thank the reviewer for the careful reading and constructive feedback. We are glad that the reviewer found the fully empirical bound and the least-favorable-distribution result meaningful. We also appreciate the reviewer’s concern that the paper may read as a refinement of a recent line of work rather than a broader conceptual advance. In the revision, we will make the significance more explicit. Our main goal is not only to sharpen constants, but to resolve two structural questions left open by Kontorovich and Painsky: whether the sharpest known $\ell_\infty$ guarantee can be made fully empirical, and what distribution is least favorable for the global worst-case risk. Theorem 2.2 addresses the first question by replacing the population quantities $V^\star(p)$ and $v^\star(p)$ with directly computable plug-in quantities $\hat V^\star$ and $\hat v^\star$; Theorem 2.1 addresses the second by showing that, up to universal constants, the global worst case is already attained by the fixed two-point law $p^\star=(1/2,1/2,0,\ldots)$, rather than by an $n$-dependent or large-support construction. The intended narrative is: Theorem 2.1 identifies the global extremal profile, Theorem 2.2 gives the practically usable empirical certificate, and Theorem 2.4 explains the finer distribution-dependent regime structure behind the multinomial problem.
>
> We agree that the current draft should better explain why these results matter beyond resolving recent open questions. Uniform $\ell_\infty$ control is the natural object when one seeks simultaneous coordinate-wise reliability rather than average-case error. In particular, the prior JMLR paper already pointed to selective inference for the most frequent symbols as a concrete application, where naive marginal intervals may fail and $\ell_\infty$ control yields valid simultaneous guarantees. The contribution of Theorem 2.2 is that it turns this type of guarantee into a directly computable data-dependent certificate. We will make this downstream relevance much more explicit in the introduction and discussion.
>
> Regarding the reviewer’s request for more intuition on $S(p)$ and $M(p)$, we agree that the exposition can be improved. These quantities are not merely technical artifacts. The term $S(p)=\sup_j p^\downarrow_j \log(j+1)$ captures the sub-Gaussian part of the difficulty, while $M(p)$ captures the additional tail-complexity regime that remains necessary in sharp non-asymptotic bounds. We will add concrete intuition and examples to explain when the variance-driven term $\sqrt{S(p)/n}$ dominates and when the tail term $M(p)/n$ becomes essential. This is precisely why Theorem 2.4 is presented as structural rather than operational: $S(p)$ and $M(p)$ explain the regime picture, whereas Theorem 2.2 is the bound intended for direct use from a single sample.
>
> We also appreciate the reviewer’s comments on soundness and presentation. In the revision, we will more clearly separate what is genuinely new from what is obtained by combining prior decoupling and local Glivenko-Cantelli ingredients, and we will explain the proof dependencies more transparently in the main text so that the role of the auxiliary reductions is easier to follow. We will broaden the introduction and related-work discussion to situate the paper relative to Kontorovich--Painsky, the local Glivenko--Cantelli line, and the recent decoupling/correlated-coordinate results. We also agree that the current organization overweights proofs in the main body; in revision we would move some proof detail out and bring a short empirical-takeaway paragraph into the main text. We will also add more interpretation around Section 2.3 and correct the minor typographical issues noted by the reviewer.
>
> Finally, we agree that the empirical section is modest, and we will clarify this point in the revision. The experiments are intended to illustrate the finite-sample informativeness and computability of the proposed empirical bound. Their purpose is to show computability and finite-sample informativeness of the new empirical bound under the same synthetic and real-data protocol as the prior baseline. We will state this scope more explicitly and improve the discussion of what the empirical comparisons do and do not show. Overall, we are grateful for the reviewer’s positive assessment of the paper’s technical core, and we will revise the paper to make the broader significance, practical interpretation, and novelty of the results much clearer.

---

> > ### Author Rebuttal · Reviewer_uGKo · 2026-04-01
> >
> > Thank you for the detailed rebuttal. But many of the issues would require substantial revision of the paper, and the rebuttal mainly provides a high-level plan for how the manuscript might be improved. At this stage, I do not think it is possible to assess the quality of those future revisions with enough confidence. Therefore, I will maintain my original score.

---

> > > ### Author Response · Authors · 2026-04-03
> > >
> > > Thank you again for the clarification and for engaging carefully with our rebuttal.
> > >
> > > We would like to emphasize that, as we understand your comment, the main issue is not with the correctness or technical quality of the results themselves, but rather with how the motivation, significance, and broader framing are currently explained in the introduction and presentation of the paper. In other words, the core gap you identify seems to be largely one of exposition and positioning, rather than a flaw in the underlying theorems or proofs.
> > >
> > > For this reason, we believe the revisions needed to address your concern are meaningful but not substantive in the sense of changing the technical content of the paper. The main results, guarantees, and proofs would remain the same; what would change is that we would explain much more clearly why the problem is important, how the paper differs from and goes beyond prior work, and how the empirical certificate can be interpreted and used.
> > >
> > > If helpful, we would also be very happy to spell out even more concretely the exact changes we would make in the manuscript—for example, how we would revise the introduction, broaden the related-work discussion, add intuition for the quantities S(p) and M(p), and make the practical motivation more explicit.
> > >
> > > In light of this, we would be grateful if you would consider whether the current score might be reconsidered.

---

### Official Review · Reviewer_foDf · 2026-03-10

**Soundness:** 3
**Presentation:** 1
**Significance:** 2
**Originality:** 3
**Overall Recommendation:** 3
**Confidence:** 3

**Summary:**

The authors propose new error bounds for the maximum-likelihood estimator (MLE)
of discrete distributions under the l-inf norm. Overall, there are 3 original
contributions to the l-inf error bounds with non-trivial proofs:
1) The worst-case expected error scales like O(sqrt(n)) where n is the sample size,
by providing an explicit least-favorable distribution for sufficiently large n.
2) A high probability upper bound on the error depending only on sample statistics
and not the underlying distribution.
3) Another high probability bound that depends on the sharpness and tail behaviour
of the true distribution.
The empirical bound (2) is demonstrated to be tighter for certain distributions than
the literature with an experimental section.

**Compliance With Llm Reviewing Policy:**

Affirmed.

**Key Questions For Authors:**

1) Why is the paper titled, “improved distribution estimation” when only the MLE is
considered and simply new error bounds are shown? The estimation quality is not
really improved.
2) How can one read the experimental results? What is the computational load of
calculating the new bounds proposed? Can they be easily used to judge the
performance of an estimator in terms of optimality?
3) Is there an interpretation of the functional introduced such as V*(p), S(p) and M(p)
or are they simply technical artifacts? You mention that they “capture the interplay
between variance and tail decay” but can you be more precise?

**Limitations:**

Neither limitations, nor open questions are discussed. A more detailed comparison
and discussion with the existing bounds, especially under the experimental results
section would be highly beneficial. Also under related work, the referenced bounds
can be restated and discussed in which terms they differ from the paper’s results.
The Impact Statement is adequate.

**Strengths And Weaknesses:**

# Soundness:
The paper provides detailed and technically sophisticated proofs of its
main contributions, although in some instances it is difficult to follow the
argumentation. For example, in the proof of Theorem 2.1, the authors claim without
justification that restricting to distributions with p_i <= 1/2 is "without loss of
generality" based on a symmetry argument for Bernoulli variables. This reasoning
does not extend to their multinomial setting where probabilities must sum to one;
applying the transformation x -> 1- x multiple coordinates simultaneously would
violate the probability simplex constraint. This likely follows from the introduced
independent Z-tilde variables but the connection is not discussed.

# Significance:
Although potentially relevant for the distribution estimation problem
which has a wide theory and range of applications, it is not so clear if the presented
bounds are so useful and interpretable. However they lay groundwork for more
simplified and useful bounds.

# Presentation:
There are a handful of notational ambiguities. For example; under
thm 2.4 it is not clear what the /\ and \/ symbols mean or there are several
expectations under the proof of thm 2.1 where it is ambiguous over which
distribution they are taken. Moreover, there are references to unnamed equations or
inequalities. The paper is organizationally obtuse in general; involving a lot of
technical proofs in the main results section and no conclusion section for the
use/implications of the work, or with future directions.

# Originality:
The paper does include original results, in particular the empirical bound
(thm 2.2) which is demonstrated to be tighter than the literature. However, as I’m not
familiar with the referenced works, I am not sure about the extent of methodological
novelty presented. Overall, although containing original results, the paper would be
better suited as a technical note as is.

---

> ### Author Rebuttal · Authors · 2026-03-30
>
> We thank the reviewer for the careful reading and constructive feedback. We are encouraged that you found the contributions original and viewed Theorem 2.2 as a genuine improvement over prior empirical bounds.
>
> Regarding the soundness of Theorem 2.1, you are right: as written, this step is not valid in the multinomial setting. The coordinatewise $x \mapsto 1-x$ symmetry is a Bernoulli argument and cannot be applied directly on the simplex. In particular, the case where one coordinate has $p_1 > 1/2$ must be treated separately, and we will correct this in the revision. This does not change the theorem's conclusion. Because at most one multinomial coordinate can exceed $1/2$, we may isolate that coordinate and define a modified vector $p'$ by $p'_1 = 1-p_1$ and $p'_i = p_i$ for $i \ge 2$. Then $\sum_i p'_i < 1$, and after sorting $p'$ we have $p'_i \le 1/i$. Therefore the same maximization argument still applies, so the extremal distribution remains $(1/2,1/2,0,\dots)$ up to universal constants. We will make this correction explicit in the revised proof. This also alerted us that wording around Theorem 2.4 is imprecise in the same way, and we will revise that as well.
>
> On significance and motivation, this paper settles two structural questions left open by recent JMLR work: whether sharp population-level $\ell_\infty$ theory admits a fully empirical analogue, and whether worst-case risk is attained by a fixed least-favorable distribution independent of $n$. Theorem 2.2 answers the first via a plug-in bound in terms of $\hat V^\star$ and $\hat v^\star$, and Theorem 2.1 answers the second by identifying a fixed two-point extremal distribution. These results resolve the main structural questions for the MLE in the current countable-alphabet $\ell_\infty$ theory. This is crucial because in many modern ML pipelines, the failure mode is the largest coordinate-wise error ($\ell_\infty$), not average error. While distribution estimation is well-studied under KL and $\ell_2$ losses, the $\ell_\infty$ setting is much less understood over countable alphabets. Our computable certificate is useful for downstream tasks like top-$k$ reporting, simultaneous rank inference, and class-prevalence estimation, where poorly estimated categories pose the main risk. Theorem 2.2 provides a sample-computable certificate ensuring that when $\|\hat p-p\|_\infty$ is small, all coordinates and rankings are simultaneously reliable.
>
> This also addresses the title concern. The title refers to improved estimation guarantees for the MLE, not to a new estimator. Our results show its minimax optimality up to constants in the worst case, together with sharper instance-dependent and fully empirical guarantees. We will revise the introduction to clarify this.
>
> Regarding the experiments: the plots compare computable confidence certificates, not estimator optimality. Our bound is evaluated from $\hat p$ alone and compared to the prior empirical benchmark and an oracle reference. For each $n$, the experiment draws $R=1000$ samples, computes $\hat p$, and evaluates each bound; the oracle is the empirical $(1-\delta)$-quantile of the realized error. Our curve is obtained directly from $\hat V^\star$ and $\hat v^\star$, so once $\hat p$ is available, the bound is easy to compute by sorting the empirical masses. We will clarify that constants are proof-level and not optimized. We will also sharpen the comparison: previous empirical bounds are driven by $\hat v^\star$-type information, whereas Theorem 2.2 tracks the sharper $\hat V^\star$ tail structure, yielding the largest gains in heavy-tail regimes.
>
> On interpretability of $V^\star(p)$, $S(p)$, and $M(p)$: $V^\star(p)$ is the variance proxy weighted by the logarithmic multiplicity cost of simultaneous control; $S(p)$ captures the asymptotic sub-Gaussian regime; $M(p)$ captures the intermediate tail regime that can dominate before $\sqrt{S(p)/n}$ takes over. We will add examples and a clearer discussion of these regimes.
>
> Finally, on presentation, we agree the current version is too proof-heavy. In revision we will define $\vee$ explicitly, name unnamed equations, make expectations explicit in Theorem 2.1, expand the comparison with Kontorovich--Painsky and LGC papers, and add a conclusion paragraph summarizing practical implications and limitations. We appreciate these suggestions and believe they will improve readability.

---

> > ### Author Rebuttal · Reviewer_foDf · 2026-04-02
> >
> > Dear authors, thank you for the careful responses - the issues have been adequately addressed and we have no further comments at this stage.

---

> > > ### Author Response · Authors · 2026-04-03
> > >
> > > We thank the reviewer for the positive acknowledgment. If there are no remaining concerns, we kindly ask you to consider updating your score to reflect that the issues have been fully addressed.

---

### Official Review · Reviewer_U4ao · 2026-03-12

**Soundness:** 3
**Presentation:** 4
**Significance:** 2
**Originality:** 2
**Overall Recommendation:** 4
**Confidence:** 3

**Summary:**

This paper studies discrete distribution estimation under the $ \ell_\infty $ loss over a countable alphabet. The main claimed contributions are: (i) a minimax characterization showing that the worst-case expected risk is $ \Theta(n^{-1/2}) $, with a fixed least-favorable distribution $p^\star=(1/2,1/2,0,\dots)$ up to universal constants for all sufficiently large $n$; (ii) a fully empirical high-probability bound for $ |\hat p-p|_\infty $ in terms of the plug-in quantities $ \hat V^\star $ and $ \hat v^\star $; and (iii) a refined distribution-dependent regime picture via the quantities $S(p)$ and $M(p)$. The paper also includes experiments, following the protocol of Kontorovich and Painsky (2025), comparing the new empirical bound against prior ones on synthetic and real datasets

**Compliance With Llm Reviewing Policy:**

Affirmed.

**Key Questions For Authors:**

1) Can the authors explain more explicitly which ingredients in Theorem 2.2 are conceptually new, versus what follows by combining existing bounds and control lemmas?

2) Do the authors have a more direct downstream application where a fully empirical $ \ell_\infty $ confidence bound improves a concrete ML task, rather than mainly improving the standalone theoretical guarantee?

Comments

I liked the paper. It is clear, mathematically tasteful, and technically interesting. The fully empirical bound is the main strength, and the least-favorable distribution result is also elegant. My overall leaning is weak accept, provided the authors correct the statement-level typo in Lemma 2.6 and better explain why these results matter beyond resolving open questions from a recent paper. In particular, the paper would be stronger if it more clearly articulated the broader significance of the fully empirical bound and the minimax characterization, rather than presenting them mainly as completions of a recent line of work.

**Limitations:**

yes

**Strengths And Weaknesses:**

\textbf{Strengths}

The paper is clear and well-organized. The main results are not too hard to follow, and the paper does a good job of explaining how they relate to the open questions left by the recent work of Kontorovich and Painsky. In particular, the fully empirical bound in Theorem 2.2 is the highlight for me: replacing the unknown population quantities by observable plug-in quantities makes the bound more meaningful and directly computable from data.

I also found the minimax result elegant. Identifying the simple two-point distribution $p^\star=(1/2,1/2,0,\dots)$ as least favorable up to constants gives a clean structural answer to the extremal question.

The experiments are modest but useful. They suggest that the proposed empirical bound is often stronger than the previous empirical benchmark.

\textbf{Weaknesses}

My main concern is about significance and originality. The work is clearly interesting and technically clean, but it feels more like a careful refinement of a very recent line of results than a major conceptual leap. The paper resolves concrete open questions, but I am not convinced that the advance is broad enough for ICML. The authors also did not explain why this is so important to solve.

There is also a clear statement-level (serious) typo in Lemma 2.6, which took me some time to parse, since, as written, the claim is plainly impossible. As written, it claims a uniform positive lower bound on $S(p)$ for every distribution $p$, but the proof itself gives examples where $S(p)$ can be arbitrarily small. This looks like an inequality-direction typo rather than a deep conceptual flaw. By contrast, Lemma 2.5 is fine, though it could use minor cleanup in wording and quantification (e.g. $n \ge 2 $).

Soundness

My impression is that the core technical ideas are plausible and likely correct, and I did not see any serious conceptual problems. Still, the typo in Lemma 2.6 should definitely be corrected.

Presentation

The presentation is one of the paper's stronger aspects. The introduction is readable, the theorem statements are clear, and the paper maintains a clean theorem-proof flow. My only suggestion is that the authors could be more explicit about separating what is genuinely new from what is derived by combining prior decoupling and local Glivenko–Cantelli results.

Significance / Originality

The results are mathematically neat, and the fully empirical bound is a useful contribution.  The novelty seems real, but moderate: this feels more like a strong refinement/completion of a recent line of work than a major new direction.

---

> ### Author Rebuttal · Authors · 2026-03-30
>
> We thank the reviewer for the thoughtful comments and for pointing out that the current draft does not sufficiently emphasize the intuition, motivation, and broader significance of the problem. We agree, and we will revise the paper accordingly. We also thank the reviewer for catching the typo in Lemma 2.6; this is a statement-level inequality-direction typo, and we will correct it in the revision.
>
> We also agree that Section 2.3 currently needs better motivation, and in the revision we will move that explanation much earlier. The role of Section 2.3 is to show that this problem has a structural regime picture that goes beyond the coarse worst-case $n^{-1/2}$ rate and explains when faster, distribution-dependent behavior occurs. In particular, it explains why the difficulty of $\ell_\infty$ distribution estimation is not governed by support size alone, but by finer properties of the distribution, namely a variance-type term and a tail-complexity term. This is important because it shows that the countable-alphabet problem is not just a minor variant of classical multinomial estimation: once one works in sup-norm and allows heavy-tailed or infinite-alphabet distributions, the right notion of difficulty becomes genuinely distribution-dependent. We will revise Section 2.3 to make this purpose much clearer and to better separate what is inherited from the local Glivenko--Cantelli framework from what is new in the multinomial/simplex-constrained setting.
>
> The main motivation for this paper is that many modern ML pipelines are sensitive to worst-case coordinate errors, making the $\ell_\infty$ metric the natural notion of performance. For a more direct downstream application, our $\ell_\infty$ guarantee provides exactly the kind of uniform, coordinate-wise control that is useful in several inference tasks. In settings such as selective or top-$k$ reporting, simultaneous rank inference, and class-prevalence estimation under distribution shift, the main failure mode is often that a small number of categories are estimated particularly poorly. A concrete use case is selective inference / top-$k$ event reporting, where a population-level oracle threshold is not operational but a sample-computable $\ell_\infty$ certificate is. A bound on $|\hat p - p|_\infty$ directly controls the largest such error, and therefore gives a simple sufficient condition for the reliability of downstream conclusions: if every coordinate is uniformly accurate, then rankings are more stable, top categories are less likely to be misidentified, and prevalence estimates for all classes come with a worst-case error certificate.
>
> We will also make the comparison to Kontorovich and Painsky (2025) more explicit. Their paper already gave the sharper population-level $V^\star$ guarantee and a fully empirical but looser $v^\star$-based guarantee; Theorem 2.2 closes this gap by upgrading the sharp $V^\star$ population bound into a fully empirical, data-computable guarantee. This matters not only as a technical completion, but because it turns a population-dependent oracle statement into a confidence bound that can actually be computed from the observed histogram alone. More broadly, we view this result as turning $\ell_\infty$ estimation from a population-level theoretical guarantee into a practically usable tool with explicit, data-driven certificates

---

> > ### Author Rebuttal · Reviewer_U4ao · 2026-03-31
> >
> > Thank you for your clarifications. I will keep my score. Best of luck!

---

### Official Review · Reviewer_tV94 · 2026-03-19

**Soundness:** 3
**Presentation:** 2
**Significance:** 3
**Originality:** 2
**Overall Recommendation:** 4
**Confidence:** 2

**Summary:**

This paper investigates discrete distribution estimation under the $\ell_\infty$ loss over countable infinite alphabets, using the framework introduced by Kontorovich and Painsky (2025). The primary contributions are: (i) establishing a global least-favorable distribution for the worst-case expected $\ell_\infty$ risk (up to a constant), which demonstrates that the global minimax rate is achieved (up to a constant) by the two-point distribution and (ii) deriving a fully data dependent, high-probability bound that substitutes the population quantities measuring the variance and complexity with their empirical estimators.

**Compliance With Llm Reviewing Policy:**

Affirmed.

**Final Justification:**

The rebuttal addressed some of my concerns.

**Key Questions For Authors:**

1. It would be helpful if the authors could state more explicitly how Theorem 2.2 improves on the earlier empirical theorems of Kontorovich and Painsky (2025), and whether a detailed comparison can be given.
2. Theorem 2.4 is interesting from a population-level perspective. Do the authors believe it can be strengthened to a fully empirical bound, similar to Theorem 2.2?

**Limitations:**

Yes

**Strengths And Weaknesses:**

Strength:
The paper provides some improvements of the 2025 JMLR paper. I find Theorem 2.1 conceptually appealing: if correct, it clarifies that the hardest instance is essentially Bernoulli.

Weakness:
The paper is technically strong, but the presentation is quite dense and difficult to follow. In particular, the development of the new quantities and bounds is highly technical, while the intuition, motivation, and broader practical implications are not sufficiently emphasized. Specifically, Section 2.3. is lack of discussion. Both the motivation and implication are unclear. Also it is not fully explained how much of Theorem 2.4 is a genuinely new multinomial contribution versus a transfer of known local Glivenko–Cantelli results.

---

> ### Author Rebuttal · Authors · 2026-03-29
>
> We thank the reviewer for the thoughtful comments and for pointing out that the current draft does not sufficiently emphasize the intuition, motivation, and broader implications of the results. We agree, and we will revise the paper accordingly.
>
> Our main motivation is that most prior work on discrete distribution estimation focuses on $\ell_1, \ell_2$ or KL-type losses, whereas in many modern ML settings the relevant quantity is the largest coordinate-wise error, making $\ell_{\infty}$ the natural metric. This is particularly important in applications that require simultaneous uniform control over all events, such as calibration, selective inference, anomaly detection, and decision-triggering pipelines, where a single badly estimated probability can dominate performance. The countable/infinite-alphabet regime is especially relevant in modern heavy-tailed settings, where bounds that scale with support size are often not informative.
>
> We will also make the comparison to Kontorovich and Painsky (2025) more explicit. Kontorovich–Painsky already gave the sharper population-level $V^\star$ guarantee and a fully empirical but looser $v^\star$-based guarantee; Theorem 2.2 closes exactly that gap by giving a fully empirical analogue at the sharper $V^\star$ level. This is important in practice because it yields a confidence bound that can be computed directly from the observed data, without access to unknown distributional parameters.
>
> We also agree that Section 2.3 currently lacks sufficient discussion. Our goal there is to show how sharp local Glivenko–Cantelli results can be transferred to the multinomial/simplex-constrained setting, and to clarify the role of the two structural quantities $S(p)$ and $M(p)$. In particular, this section explains that beyond the worst-case $n^{-1/2}$ regime, the effective difficulty is governed by finer properties of the distribution, namely a variance-type term and a tail-complexity term, rather than by support size alone. We will revise this section to better explain what is new in the multinomial setting and what is inherited from the LGC framework.
>
> Regarding empirical evidence: we already include experiments showing that the new fully empirical bound improves (on real-world datasets) over the previous fully empirical bound of Kontorovich and Painsky (2025, Theorem 4), and we agree that this point should be highlighted more clearly. In the revision, we will expand this discussion and add further experiments emphasizing this empirical improvement on additional real-world style distributions/applications.
>
> Finally, we believe that a sharper empirical analogue of Theorem 2.4 is plausible, and pursuing such a result is an interesting direction for future work.

---

> > ### Author Rebuttal · Reviewer_tV94 · 2026-04-02
> >
> > Thank you for your response. I will slightly increase my score from 3 to 4.

---

### Decision · Program_Chairs · 2026-04-30

**Decision:**

Accept (regular)

**Comment:**

This paper analyzes the $\\ell_\\infty$ error when estimating a discrete distribution with the maximum likelihood estimator. The theory extends prior work by constructively identifying the worst-case distribution for minimax analysis and giving fully-empirical finite-sample high-probability bounds on the error. This contribution is sound and worthwhile, but somewhat incremental.

Perhaps the more interesting space of contribution is that of the distribution-dependent behavior of the $\\ell_\\infty$ error. Here, however, the results could benefit from significant improvement in presentation and interpretation. For instance, it appears that the $1/\\sqrt{n}$ behavior will dominate asymptotically with only brief periods of fast convergence at the $1/n$ rate, but the characterization that is offered (through bounding the $M$ and $S$ functionals of the distribution) does not give a clear idea of how this  transition from one mode to the other occurs for specific distributions. The reader thus walks away with only a vague understanding of how exactly the error's behavior depends on the distribution, which is the whole point.

Thus, while the paper is an appreciable theoretical contribution to discrete distribution estimation, it could benefit from a more coherent development and presentation of the distribution-dependent results, to make the narrative of the work more clear and actionable. Several technical clarifications were made during the discussion phase, and these should also be incorporated into the revision.